# Arginine methylation of METTL14 promotes RNA $N^6$-methyladenosine modification and endoderm differentiation of mouse embryonic stem cells

Xiaona Liu[1,2,3,9], Hailong Wang[4,9], Xueya Zhao[2,9], Qizhi Luo[5,9], Qingwen Wang[2], Kaifen Tan[2], Zihan Wang[2], Jia Jiang[2], Jinru Cui[2], Enhui Du[2], Linjian Xia[2], Wenyi Du[6], Dahua Chen[7,3✉], Laixin Xia [2,8✉] & Shan Xiao [2✉]

RNA $N^6$-methyladenosine (m$^6$A), the most abundant internal modification of mRNAs, plays key roles in human development and health. Post-translational methylation of proteins is often critical for the dynamic regulation of enzymatic activity. However, the role of methylation of the core methyltransferase METTL3/METTL14 in m$^6$A regulation remains elusive. We find by mass spectrometry that METTL14 arginine 255 (R255) is methylated (R255me). Global mRNA m$^6$A levels are greatly decreased in METTL14 R255K mutant mouse embryonic stem cells (mESCs). We further find that R255me greatly enhances the interaction of METTL3/METTL14 with WTAP and promotes the binding of the complex to substrate RNA. We show that protein arginine N-methyltransferases 1 (PRMT1) interacts with and methylates METTL14 at R255, and consistent with this, loss of PRMT1 reduces mRNA m$^6$A modification globally. Lastly, we find that loss of R255me preferentially affects endoderm differentiation in mESCs. Collectively, our findings show that arginine methylation of METTL14 stabilizes the binding of the m$^6$A methyltransferase complex to its substrate RNA, thereby promoting global m$^6$A modification and mESC endoderm differentiation. This work highlights the crosstalk between protein methylation and RNA methylation in gene expression.

[1] School of Life Sciences, University of Science and Technology of China, Hefei, P.R. China. [2] Department of Developmental Biology, School of Basic Medical Sciences, Southern Medical University, Guangzhou, P.R. China. [3] State Key Laboratory of Reproductive Biology, Institute of Zoology, Chinese Academy of Sciences, Beijing, China. [4] Kingmed School of Laboratory Medicine, Guangzhou Medical University, Guangzhou, P.R. China. [5] Department of Forensic Toxicology, School of Forensic Medicine, Southern Medical University, Guangzhou, P.R. China. [6] Sichuan MoDe Technology Co., Ltd, Chengdu, P. R. China. [7] Institute of Biomedical Research, Yunnan University, Kunming, P.R. China. [8] State Key Laboratory of Organ Failure Research, Southern Medical University, Guangzhou, P.R. China. [9] These authors contributed equally: Xiaona Liu, Hailong Wang, Xueya Zhao, Qizhi Luo. ✉email: chendh@ioz.ac.cn; xialx@smu.edu.cn; asdfg@smu.edu.cn

$N$[6]-methyladenosine (m6A) is the most abundant mRNA modification in mammals, and it has been widely explored following the development of the methylated RNA immunoprecipitation-sequencing (MeRIP-seq) method[1,2]. mRNA m6A is mostly found in stop codons, 3′untranslated regions (3′UTRs), and long exons that possess a consensus motif of RRACH (R means G or A, and H means A, C or U)[1,3]. m6A is a dynamically reversible modification, which is synergistically regulated by the methyltransferase complex METTL3-METTL14-WTAP[4] and demethylases FTO and ALKBH5[5–7]. m6A plays important roles in RNA stability[8,9], splicing[10,11], translation[9,12], and other cellular processes[13–16]. Dysregulation of m6A modification has been shown to result in impairment of stem cell self-renewal or differentiation[17–22], abnormal developmental process[6,23–26], and tumorigenesis[27,28] in animals.

Protein arginine methylation is a key post-translational modification (PTM) catalyzed in large part by nine protein arginine methyltransferases (PRMTs)[29]. PRMTs are divided into three categories: type I PRMTs catalyze mono-methylation and asymmetric di-methylation; type II PRMTs catalyze mono-methylation and symmetric di-methylation, and type III PRMTs only catalyze mono-methylation[30,31]. PRMT1 is the predominant type I arginine methyltransferase and is responsible for 85% of arginine methyltransferase activity in mammals[32,33]. Arginine methylation of proteins can regulate their activities through modulating their binding interactions[29] with proteins and nucleic acids, especially RNAs[34–36]. It has also been shown to be involved in biological processes such as gene transcription[37–39], RNA splicing[40,41], cell signaling[42], cell fate decision[43,44], and other cellular processes[37,45]. Although phosphorylation of METTL14[46] and SUMOylation of METTL3[47] have been detected, arginine methylation of m6A methyltransferase complex has not been reported. Moreover, the role of PTMs on the activity of the methyltransferase complex has not been fully characterized.

Here, we report arginine methylation at position arginine 255 (R255me) of METTL14, identified using liquid chromatography tandem mass spectrometry (LC–MS/MS). Methylation of METTL14 at R255 regulates the activity of the m6A methyltransferase complex and enhances global RNA m6A modification. We show that PRMT1 contributes to R255 methylation, and loss of R255me affects the endoderm differentiation of mouse embryonic stem cells (mESCs). This study elucidates a critical role for arginine methylation in RNA m6A regulation and mESC differentiation, suggesting a direct link between protein methylation and RNA methylation.

## Results

### METTL14 is methylated at arginine 255

To investigate the methyl-arginine status of METTL14, we generated a HeLa cell line stably expressing S-Flag-SBP (SFB)-tagged METTL14, which we confirmed has a similar expression level as endogenous METTL14 (Supplementary Fig. 1a). Western blots showed that immunoprecipitated SFB-METTL14 protein was arginine methylated (Fig. 1a). This methylation was remarkably decreased by treatment with an arginine methylation inhibitor, periodate oxidized adenosine (AdOx) (Fig. 1a). Arginine methylation of METTL14 was also detected in E14Tg2a mouse embryonic stem cells (mESCs) (Supplementary Fig. 1b). To determine the methylation sites, SFB-METTL14 was purified by tandem affinity purification (TAP) (Supplementary Fig. 1c) and subjected to LC–MS/MS. A reported phosphorylation site at S399[46] was found, suggesting that our approach captured bona fide protein modifications. We found the R255 site of METTL14 was mono-methylated (Fig. 1b, Supplementary Fig. 1d). To confirm the methylation at R255, we mutated R255 to lysine (R255K), and

found the mono-methylation of immunoprecipitated METTL14 was greatly reduced (Fig. 1c). Alignment of METTL14 proteins from various species revealed that R255 is conserved from humans to yeast (Supplementary Fig. 1e). Based on the crystal structure of METTL14[48], R255 sits at the interface between METTL3 and METTL14 and is at the surface of the substrate RNA-binding site (Fig. 1d, Protein Data Bank ID code 5IL1, [https://doi.org/10.2210/pdb5IL1/pdb]), indicating that the R255 methylation of METTL14 may be functionally important.

### METTL14 R255K decreases mRNA m6A modification

To investigate the function of METTL14 R255 methylation, we generated a METTL14 R255K cell line in mESCs using CRISPR-Cas9 (Supplementary Fig. 2a, b). The protein level of METTL14 was not substantially affected by this amino acid substitution (Fig.2a). Dot blots showed hypo-methylation of m6A in mRNA isolated from the R255K cell line (Fig. 2b), and the m6A/A ratio of mRNA was significantly reduced in R255K cells compared to wild-type (WT) cells, as assessed by LC–MS/MS (Fig. 2c, Supplementary Fig. 2c). Overexpression of METTL14 rescued this m6A decrease (Fig. 2c, Supplementary Fig. 2d).

We then performed MeRIP-seq from WT and METTL14 R255K cells to explore the genome-wide effect of R255 methylation. Consistent with previous reports[1,3], m6A modifications in both WT and R255K cells mostly occurred in coding regions and 3′UTRs (Fig. 2d, Supplementary Fig. 2e) at the consensus motif GG(m6A)C (Supplementary Fig. 2f). However, global hypo-methylation of m6A was seen in the transcriptome in the R255K mutant (Figs. 2d, e). The number of m6A peaks on RNA decreased from 17,693 m6A peaks in the WT cells to 9437 peaks in R255K cell line. MeRIP-qPCR confirmed the decreased m6A levels on individual genes in R255K cells (Fig. 2f). 5008 m6A regions were significantly decreased in the R255K mutant. These m6A regions were enriched for an A(G/A)AC motif, while the constitutive m6As were enriched for a GGAC motif (Supplementary Fig.2g). Previous reports suggest that the A(G/A)AC motif may not be an ideal substrate for METTL14[48]. Not surprisingly, we found that the R255 dependent m6As are less abundant (Fig. 2g). Lower abundance m6A peaks are often observed in highly expressed transcripts[2]. Consistent with this, the expression levels of genes associated with these R255 regulated m6A regions were significantly higher than the other genes in WT cells (Fig. 2h). Together, our results indicate that loss of METTL14 R255 methylation greatly reduces RNA m6A modification.

### PRMT1 physically interacts with and methylates METTL14

Arginine mono-methylation is catalyzed by PRMTs[29,37]. To identify the PRMT responsible for METTL14 R255 methylation, proteins that interacted with METTL14 were pulled down by TAP and detected using LC–MS/MS in HeLa cells (Supplementary Fig. 3a). METTL3 and WTAP, two components of the m6A methyltransferase complex (Fig. 3a) were captured, indicating the pull down assay captured biologically relevant interacting partners. PRMT1 was found abundantly in the pulled down protein pools (Fig. 3a). The interaction between PRMT1 and METTL14 was confirmed by co-immunoprecipitation (Co-IP) (Figs. 3b, c, and Supplementary Fig. 3b–e). We next sought to determine the domain mediating the interaction. Three truncated METTL14 fragments (Fig. 3d) with deletion of RGG repeats (aa 396–456), nuclear localization signals (NLS, aa 1–146), or the methyltransferase domain (MTD, aa 147–395) were transfected into HeLa cells with PRMT1, and Co-IP was carried out (Fig. 3d). The METTL14 RGG repeats in the carboxyl terminal were essential for the interaction between METTL14 and PRMT1 (Fig. 3e).

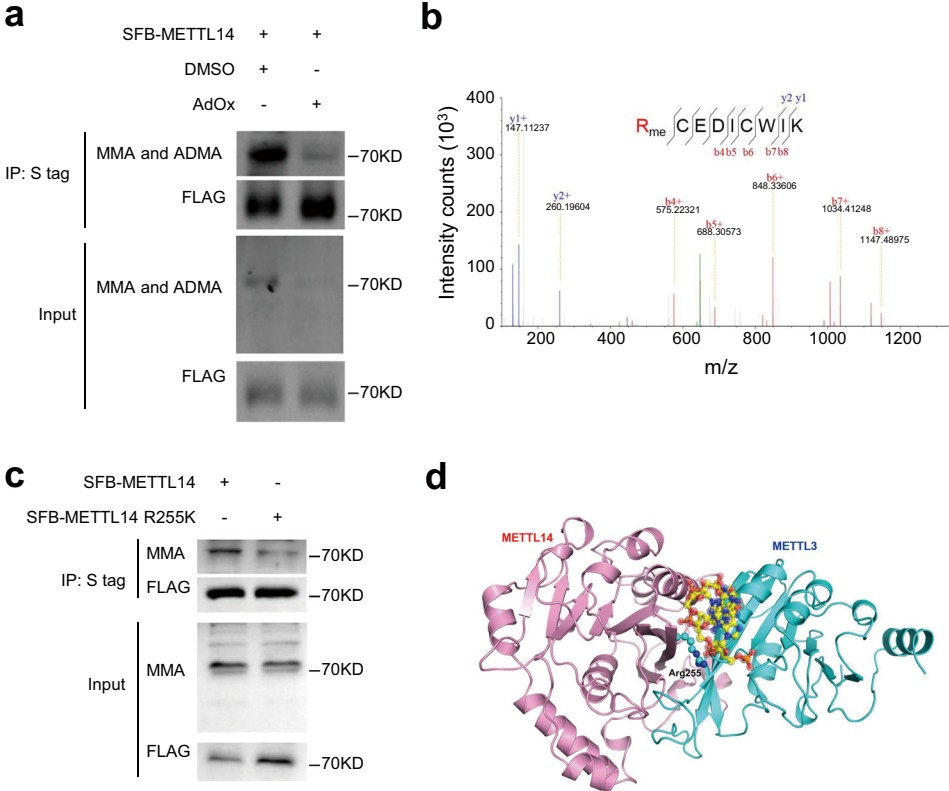

**Fig. 1 METTL14 is methylated at arginine 255. a** Arginine methylation of immunoprecipitated S-Flag-SBP (SFB) tagged METTL14 protein detected by western blot in HeLa cells. MMA mono-methyl arginine, ADMA asymmetric dimethyl arginine. The experiment was performed once in HeLa cells. **b** Mass spectrometry showing that METTL14 R255 is methylated. The mass of each b ion and y ion are shown. The mass error of the peptide identification is 0.26326 ppm. **c** Arginine mono-methylation of S-Flag-SBP (SFB) tagged METTL14 and METTL14 R255K proteins analyzed by western blot. Representative figures of three independent replicates are shown. Source data for (**a**) and (**c**) are provided as a Source Data file. **d** Crystal structure ribbon diagram indicating the position of METTL14 R255 in the METTL3 (cyan) and METTL14 (pink) heterodimer. RNA is shown as a ball-and-stick model in the middle.

To further confirm R255 was methylated by PRMT1, we carried out an in vitro arginine methylation assay. Three types of peptides, R255 mono-methylated, R255 WT, and R255K mutated peptides, were incubated with PRMT1 protein and methyl donor S-Adenosyl-L-methionine (SAM) and then blotted on a membrane and detected with a mono-methylation antibody. We found that R255 WT peptides were methylated by PRMT1, while R255K peptides were methylated to a lesser extent (Fig. 3f). This residual mono-methylation signal may be due to the presence of other arginine sites in the peptides. To reduce the background signal caused by other potential methylated sites, we generated truncated METTL14 and METTL14 R255K (METTL14-T and R255K-T) and transfected these constructs into cells to see if PRMT1 could methylate R255. We found that loss of PRMT1 reduced the mono-methylation of the METTL14-T construct but had little effect on the R255K-T construct (Supplementary Fig. 3f). Together, these results indicate that PRMT1 interacts with and methylates METTL14 at R255.

**PRMT1 regulates mRNA m⁶A modification**. Considering that PRMT1 methylated METTL14 R255, we investigated the relationship between PRMT1 and RNA m⁶A modification. We found that PRMT1 overexpression increased mRNA m⁶A in HeLa and mESCs (Figs. 4a, e, and Supplementary Fig. 4a), while PRMT1 knockdown decreased m⁶A (Fig. 4b, Supplementary Fig. 4b). Next, we generated PRMT1 mutant mES cell lines using CRISPR-Cas9 (Supplementary Fig. 4c) and confirmed by western blot that the PRMT1 protein was not detectable in the mutant cells (Supplementary Fig. 4d). LC–MS/MS data showed that the m⁶A

level of mRNA decreased in PRMT1 mutant cells (Fig. 4c, Supplementary Fig. 4e), an effect that could be rescued by PRMT1 overexpression (Fig. 4c, Supplementary Fig. 4f). Accordingly, gene-specific MeRIP-qPCR showed that the genes whose m⁶A level decreased in the METTL14 R255K cell line also had decreased RNA m⁶A levels in PRMT1 mutant cells (Fig. 4d). Moreover, overexpression of PRMT1 in R255K mESCs could not rescue the decreased mRNA m⁶A level (Fig. 4e). These results indicate that PRMT1 may regulate RNA m⁶A deposition through METTL14 R255 methylation.

**METTL14 R255 methylation stabilizes the methyltransferase complex to its substrate RNA**. The m⁶A methyltransferase complex consists of three components: WTAP, METTL3, and METTL14. METTL3 is the catalytic subunit, METTL14 structurally supports METTL3 and recognizes RNA substrates[48], and WTAP is required for the recruitment of METTL3-METTL14[49]. To dissect the mechanism giving rise to the decrease in RNA m⁶A in the METTL14 R255K cell line, we looked into the properties of the methyltransferase complex. The protein level of METTL3 was not changed in R255K mESCs, whereas the amount of WTAP increased (Fig. 5a, Supplementary Fig. 5a). Then we separated the cytoplasmic and nuclear proteins, and western blots showed that subcellular localization of these three methyltransferase complex components in the R255K cell line were comparable to that in WT cells (Fig. 5b, Supplementary Fig. 5a). Next, we investigated the stability of the methyltransferase complex. Compared to WT METTL14, the association between METTL14 R255K and METTL3 did not change (Figs. 5c, d, Supplementary Fig. 5b), but

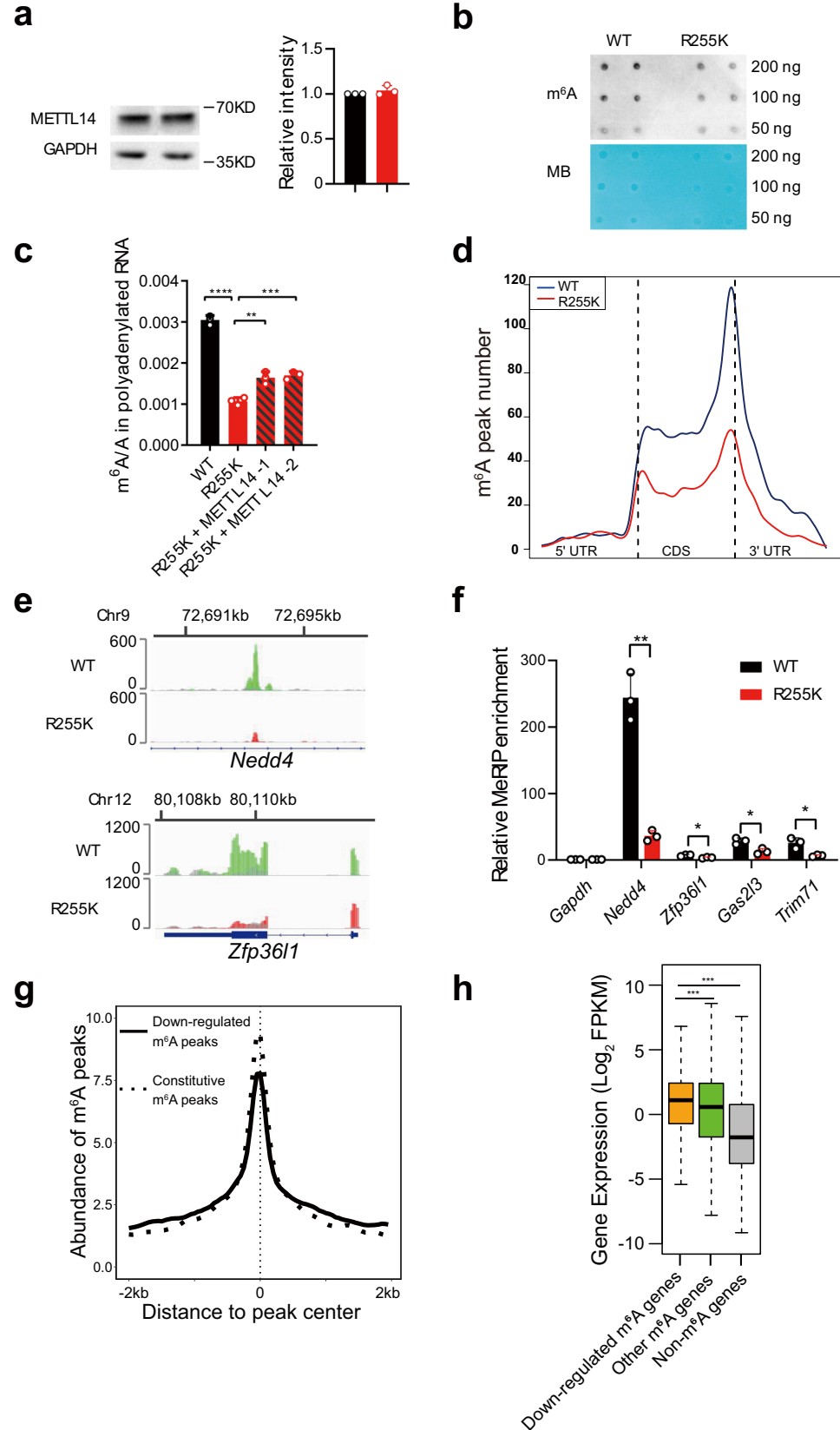

the association between METTL14 R255K and WTAP greatly decreased (Fig. 5d, Supplementary Fig. 5b and c). This suggests the increase of WTAP protein in the R255K cells may be an attempt to compensate for the decreased interaction between WTAP with METTL14-METTL3.

Given that R255 is positioned near the site of METTL14 RNA binding, we simulated the binding of METTL14 R255me with RNA substrates using molecular docking and dynamics simulations. We generated the initial structure of METTL14 based on the crystal structure of the METTL3-METTL14 complex (PDB ID

**Fig. 2 METTL14 R255K decreases mRNA m$^6$A modification. a** Western blot showing the protein level of METTL14 in WT and METTL14 R255K knock-in mESCs. Representative figures of three blots are shown. The band intensity of METTL14 relative to GAPDH is shown in the right-hand panel, and data are mean ± s.d. from three independent replicates. **b** Dot blot showing the m$^6$A mRNA modification levels from WT and METTL14 R255K mESCs. Methylene blue (MB) was used as a loading control. The experiment was performed once. **c** LC–MS/MS quantification of m$^6$A abundance in mRNA from WT, R255K, or R255K with METTL14 overexpressed cells. Data are mean ± s.d. of three independent experiments. Two-sided Student's *t* test. **d** Distribution of m$^6$A peaks from WT and METTL14 R255K cells in transcriptomic regions. **e** Profile of m$^6$A distribution on *Nedd4* and *Zfp36l1* in WT and R255K cell lines. Shown is normalized density of IP (green or red) versus input (gray). Values on the left show the range of IP and input density. **f** MeRIP-qPCR verifying decreased m$^6$A peaks on *Nedd4*, *Zfp36l1*, *Gas2l3*, and *Trim71*. Enrichment of MeRIP versus input RNA was normalized against *Gapdh*. Data are mean ± s.d. from three independent experiments. Two-sided Student's *t* test. **g** Abundance of m$^6$A signals around the m$^6$A peak center. Solid line, downregulated m$^6$A peaks in R255K. Dotted line, constitutive m$^6$A peaks. **h** Box plot showing the expression levels of genes with decreased m$^6$A abundance in R255K cells, other m$^6$A modified genes, and non-m$^6$A modified genes. The center line indicates median value, and the box and whiskers represent the interquartile range (IQR) and 1.5 IQR, respectively. Two-sided Mann–Whitney *U* test. Source data for (**a**), (**c**), (**f**), and (**h**) are provided as a Source Data file.

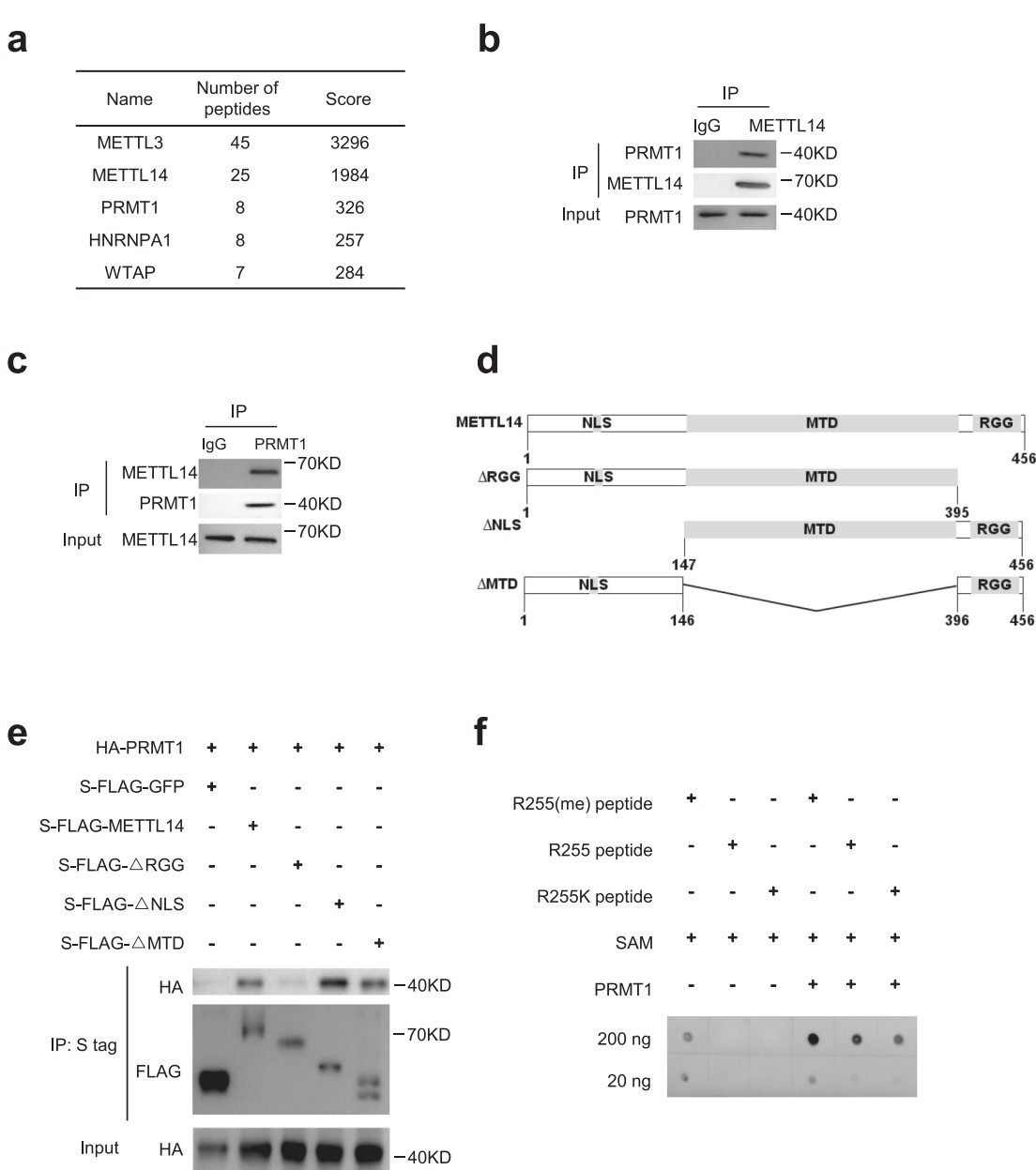

**Fig. 3 PRMT1 physically interacts with and methylates METTL14. a** List of the five most abundant proteins identified in the tandem affinity purification (TAP) pull down assay with METTL14 identified by LC–MS/MS using Mascot search engine in HeLa cells. **b–c** Co-IP of endogenous METTL14 (**b**) and PRMT1 (**c**) in HeLa cells. **d** Schematic of METTL14 truncations. MTD methyltransferase domain, NLS nuclear localization signal, RGG RGG repeats. **e** Western blot showing interactions of PRMT1 and ectopically expressed full-length or truncated METTL14 in HeLa cells. Representative figures of three independent replicates are shown. **f** In vitro methylation of R255me, R255, and R255K peptides by PRMT1. Representative figures of two independent replicates are shown. SAM S-Adenosyl-L-Methionine. Source data for (**b**), (**c**), (**e**) and (**f**) are provided as a Source Data file.

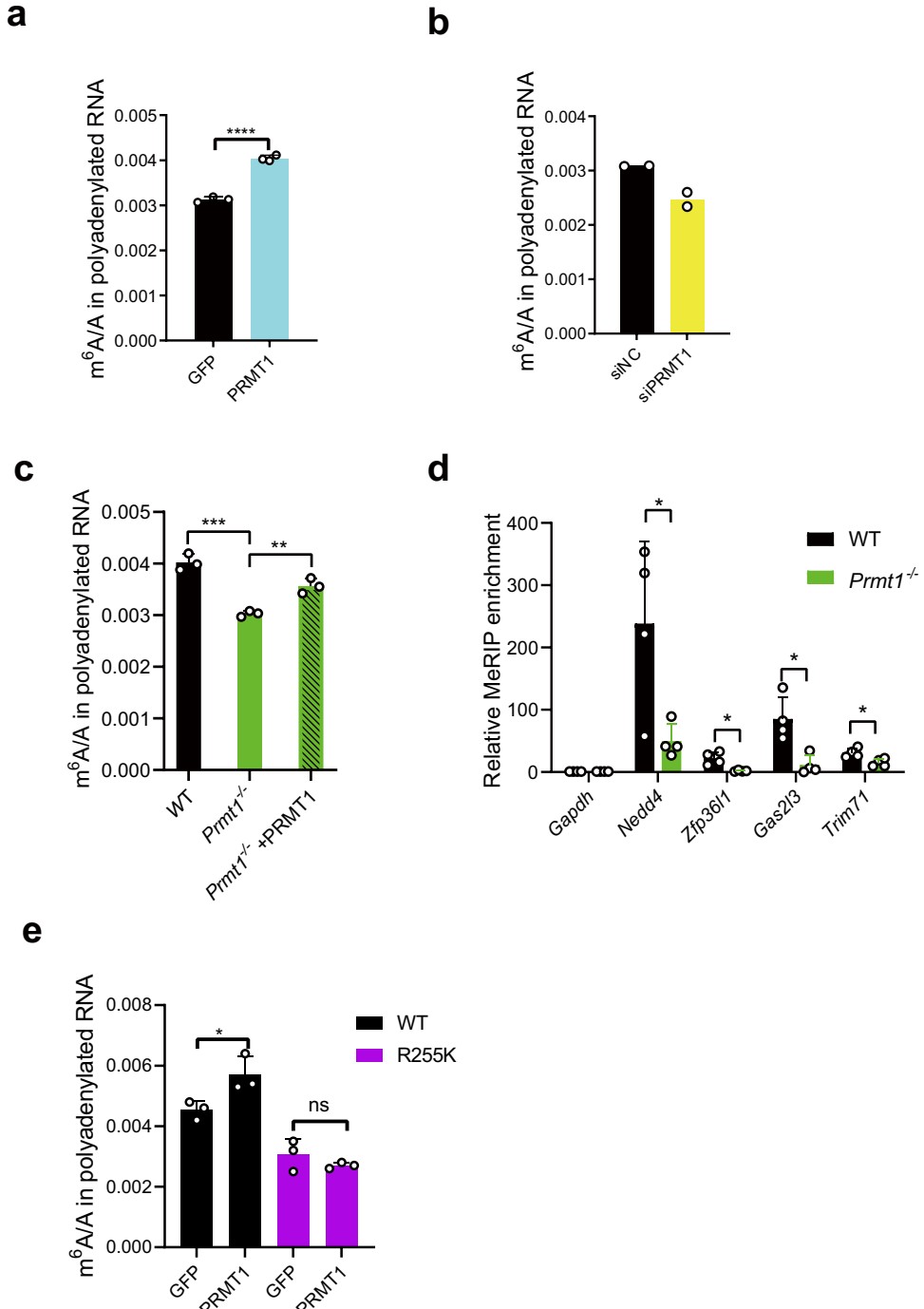

**Fig. 4 PRMT1 regulates mRNA m⁶A modification. a** m⁶A level of mRNA in HeLa cells overexpressing HA-tagged PRMT1 or GFP control. Data are mean ± s.d. from three independent experiments. Two-sided Student's *t*-test. **b** m⁶A level of mRNA in HeLa cells after siRNA-mediated knockdown of PRMT1. Data from two independent experiments are shown. **c** m⁶A level of mRNA in WT, *Prmt1⁻/⁻*, and PRMT1 overexpressed *Prmt1⁻/⁻* mESCs. Data are mean ± s.d. from three independent experiments. Two-sided Student's *t*-test. **d** MeRIP-qPCR analysis of m⁶A peaks on indicated genes in WT and *Prmt1⁻/⁻* mESCs. Data are mean ± s.d. from four independent experiments. Two-sided Student's *t* test. **e** m⁶A level of mRNA in WT and METTL14 R255K mESCs after the overexpression of GFP or PRMT1. Data are mean ± s.d. from three independent experiments. Two-sided Student's *t*-test. Source data for (**a–e**) are provided as a Source Data file.

5IL1)[48]. The RNA structure of GGACU was docked onto the RNA binding region of METTL14[48]. We next simulated the molecular dynamics (MD) of RNA with R255 unmethylated or mono-methylated METTL14[50,51]. The root mean square fluctuation (RMSF) of RNA in the R255 methylated complex was lower than that of the unmethylated complex, suggesting that the

structure of RNA is more stable in the presence of R255me (Supplementary Fig.6e). We measured the hydrogen bond interaction and binding energy of RNA with METTL3-METTL14 and found that after R255 mono-methylation, the average number of hydrogen bonds increased from 4.95 to 8.09 (Supplementary Fig. 6f), and the binding energy decreased from

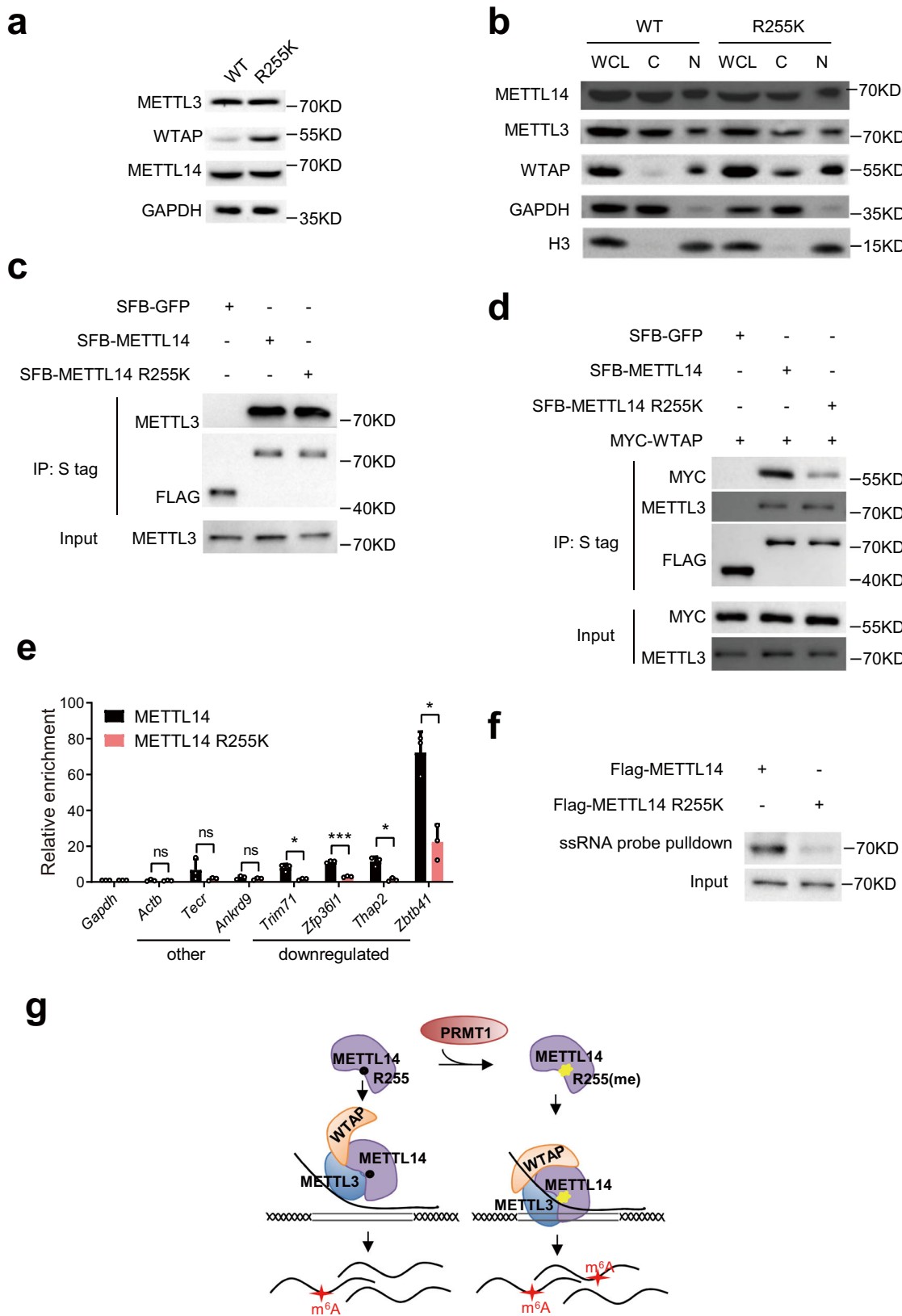

−167.478 ± 142.476 kJ/mol to −625.963 ± 169.704 kJ/mol after 30 ns simulation (Supplementary Fig. 6g), indicating the binding of RNA with METTL3-METTL14 was more stable after R255 mono-methylation. Then we tested if the RNA-binding ability of METTL14 is affected by R255me using RNA photoactivatable-ribonucleoside-enhanced crosslinking and immunoprecipitation

qPCR. We found that the METTL14 R255K exhibited reduced RNA-binding ability for transcripts with decreased m6A modification in R255K cells (Fig. 5e). A pull-down assay was conducted using a single-stranded RNA probe containing the m6A motif RRACH. Less METTL14 R255K protein was pulled down compared with WT METTL14, suggesting that METTL14

**Fig. 5 METTL14 R255 methylation stabilizes the methyltransferase complex to its substrate RNA. a** Western blot of methyltransferase complex components METTL3, METTL14 and WTAP proteins in WT and METTL14 R255K mESCs. Representative figures of two independent replicates are shown. **b** Western blot showing the subcellular location of METTL3, METTL14, and WTAP proteins in WT and METTL14 R255K mESCs. WCL whole cell lysate, C cytoplasmic fraction, N nuclear fraction. Representative figures of two independent replicates are shown. **c** Co-IP showing the interaction of METTL3 with SFB-tagged METTL14 or METTL14 R255K in HeLa cells. Representative figures of four independent replicates are shown. **d** Co-IP showing the interaction of MYC-WTAP and METTL3 with SFB tagged METTL14 or METTL14 R255K. The experiment was performed once in HeLa cells. SFB-GFP served as a control in (**c**) and (**d**). **e** qPCR showing the binding between METTL14 or METTL14 R255K and RNA with m$^6$A downregulated or other m$^6$A peaks. Enrichment was normalized against *Gapdh*. Data are mean ± s.d. from three independent experiments. Two-sided Student's *t*-test. **f** Western blot showing METTL14 or METTL14 R255K protein pulled down by ssRNA probe containing RRACH domains. Representative figures of three independent replicates are shown. Source data for (**a**–**f**) provided as a Source Data file. **g** Proposed model showing the mechanism of METTL14 R255 methylation affecting the deposition of m$^6$A methylation on mRNA. R255, arginine 255; R255(me), methylated arginine 255.

R255K has weaker RNA-binding affinity (Fig. 5f). Moreover, we found that R255me changed the conformation of the arginine side-chain, leading to a change in the RNA binding pattern on the protein surface (Supplementary Fig. 6h). The 5′ phosphate group of RNA turned from His401 to R471 after R255 mono-methylation (Supplementary Fig. 6i).

To determine whether PRMT1 affects the binding of METTL14 with METTL3 and WTAP, we knocked down PRMT1 in HeLa cells and found that the binding between METTL14 and WTAP was weaker, while the binding between METTL14 and METTL3 was relatively unaffected (Supplementary Fig. 5d). The changes of the binding between METTL14 and METTL3/WTAP in PRMT1 overexpression cells were not detected (Supplementary Fig. 5e). Altogether, these results show that METTL14 R255 methylation stabilizes the methyltransferase complex and enhances the RNA binding ability of METTL14 (Fig. 5g).

**Methylation of METTL14 at arginine 255 is required for normal mESC endoderm differentiation.** Loss of METTL14 leads to a lack of m$^6$A modification and impairment of self-renewal in mESCs[17,19,52]. To understand the biological importance of METTL14 methylation at R255, we first characterized R255K mESCs. Compared with WT cells, the R255K cell colonies were flattened and did not exhibit the typical dome shape (Supplementary Fig. 7a). Additionally, the mutant cells displayed less alkaline phosphatase (AP) positive colonies (Supplementary Fig. 7b), although the proliferation rate of R255K cells resembled WT cells (Supplementary Fig. 7c). The expression of pluripotency genes was slightly decreased in R255K cells, but the mESCs were still pluripotent with high expression of pluripotency genes and low expression of differentiation-specific genes (Supplementary Fig. 7d).

Since the loss of METTL14 methylation at R255 greatly reduced the m$^6$A level on RNA, we clustered the genes associated with downregulated m$^6$A in R255K cells according to their involvement in various biological processes and found that they were enriched for signaling pathways regulating pluripotency of stem cells and TGF-beta signaling pathways (Fig. 6a). We, therefore, checked the differentiation capacity of R255K mESCs through an in vitro embryoid body (EB) differentiation assay using a suspension culture system in the absence of leukemia inhibitory factor (LIF). Unlike the rounded spheres WT mESCs generated, the EBs of R255K mESCs were irregular at day 6 (Supplementary Fig. 7e). As differentiation progressed, ectoderm and mesoderm markers showed similar increases in R255K and WT cells. However, there was substantially less of an increase of endoderm markers in R255K cells relative to WT cells (Fig. 6b). When we directed the mESC to definitive endoderm fates using activin A and bFGF, the differentiation of R255K cells was hindered (Fig. 6c), indicating that loss of R255me impacts endoderm differentiation.

To obtain a full picture of R255 target genes that are important for mESC self-renewal or differentiation, we first assessed global RNA expression in WT and R255K mESCs using RNA-seq. We found that the expression level of most genes with R255 regulated m$^6$A peaks was not substantially changed (Supplementary Fig. 7f). For those genes that were differentially expressed, we determined they were not enriched in self-renewal or differentiation related processes (Supplementary Table 4). Given that m$^6$A can promote or inhibit targeted RNA degradation[8,9], we measured mRNA stability using RNA-seq after transcription inhibition with Actinomycin-D in WT and R255K mESCs. We found that over 67.8% of R255 targeted genes degraded slower in R255K cells than they did in WT cells (fold change of half-life >1.5, Supplementary Fig. 7g). Among these genes are *Smad6/7*, known antagonists of activin-Nodal-TGFβ signaling[53], and *Klf4*, an endoderm differentiation inhibitor[54]. We further confirmed that the m$^6$A level of these genes was decreased (Fig. 6d, Supplementary Fig. 7h), and the transcripts of these genes were more stable in R255K cells (Fig. 6e). We hypothesized that the prolonged lifetime of these genes and the network they regulated participated in the differentiation of R255K cells.

These results suggest that loss of METTL14 R255 methylation affects the normal transition of cell fates in mESCs, especially during endoderm differentiation, which may underlie the importance of R255 methylation for normal mESC development.

## Discussion

RNA m$^6$A modification has been widely appreciated as an important post-transcriptional modification, but its dynamic regulation is largely unstudied. In this study, we identified a post-translational modification of the m$^6$A writer complex component METTL14, namely methylation by PRMT1 at R255. This modification promotes global m$^6$A modification and endoderm differentiation in mESCs.

Two methyltransferases, METTL3 and METTL14, form the core catalytic complex[48] that is accompanied by additional subunits such as WTAP[49]. Structural studies have shown that METTL14 serves as an RNA-binding platform but does not catalyze methyl-group transfer[48]. Here, we show that arginine methylation of METTL14 promotes the binding of METTL14 to its RNA substrates. These results suggest an expanded role for METTL14 as a versatile and flexible scaffold to regulate interactions between the methyltransferase complex and substrate RNA. Recent studies showed that METTL14 directly binds to H3K36me3 to recruit the methyltransferase complex to mRNA 3′ UTRs[16]. It will be interesting to explore whether arginine methylation of METTL14 is also involved in this process.

Arginine methylation functions in gene expression and various biological processes[29,37]. For example, fragile X mental retardation protein (FMRP) is methylated by PRMT1 to modulate the RNA binding activity of FMRP[34,55]. The arginine methylation of HuD affected its RNA binding ability and reduces the half-life of bound

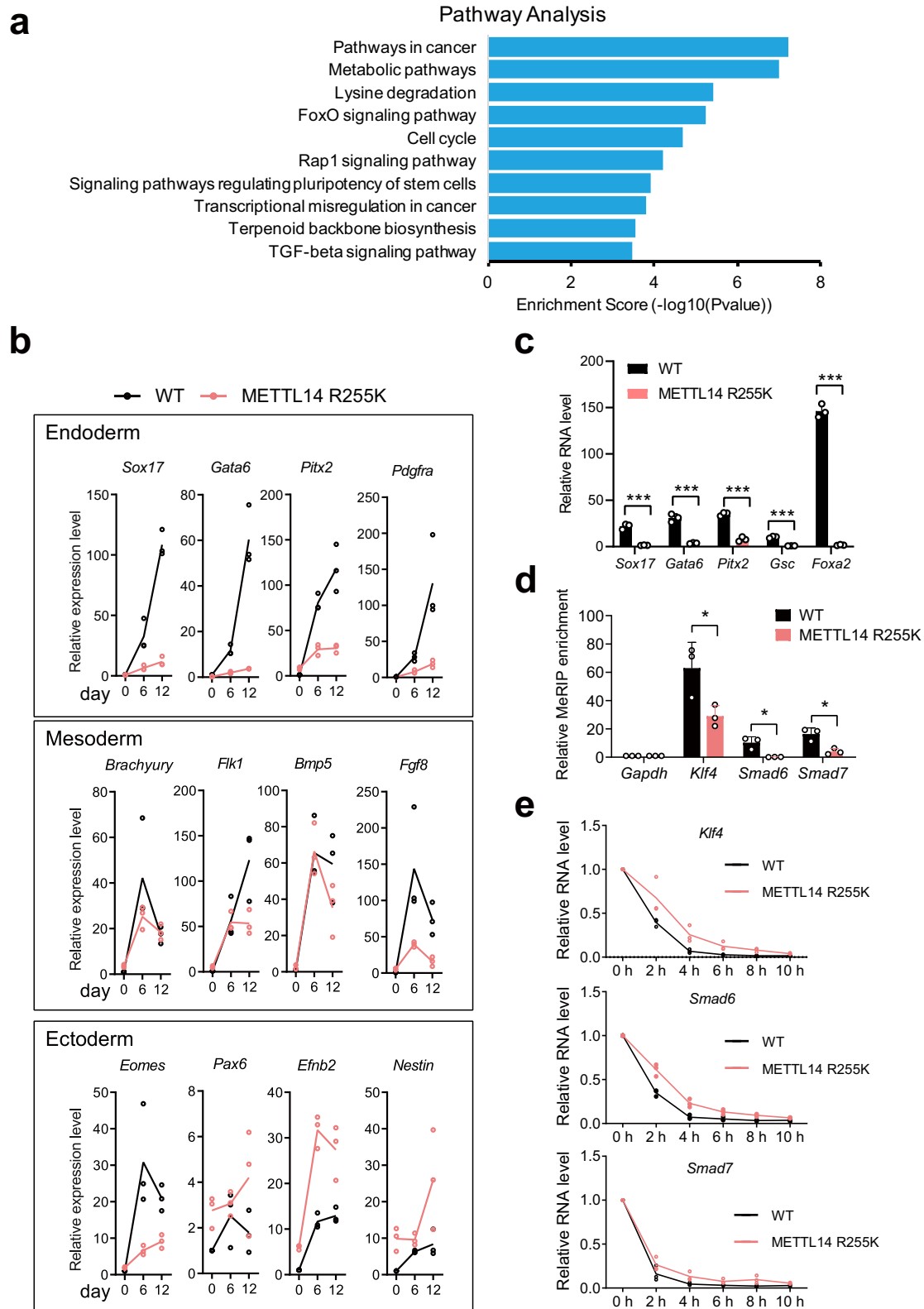

RNAs[56]. Here, we show that PRMT1 methylates METTL14 at R255 to stabilize the m[6]A methyltransferase/RNA complex. This finding is consistent with the conventional role of arginine methylation in regulating nucleic acid interactions[34,36,56]. Considering that PRMT1 knockdown has only a mild effect on global m[6]A methylation levels, we infer that other members of the PRMT family may be redundantly involved in METTL14 methylation. In the future, it will be interesting to determine context-dependent PRMT methyltransferases for METTL14 methylation in development and disease.

In summary, our findings demonstrate how arginine methylation fine tunes the regulation of m[6]A methyltransferase activity

**Fig. 6 Methylation of METTL14 at arginine 255 is required for normal mESC endoderm differentiation. a** KEGG pathway enrichment analysis of genes associated with downregulated m⁶A regions in R255K cells. One-sided hypergeometric test, *P* value was adjusted by Benjamini and Hochberg method. **b** Quantitative RT-PCR analysis of differentiation markers of ectoderm, mesoderm, and endoderm in WT and R255K mESCs. The expression level was normalized against *Gapdh* then to WT at day 0. Data from three independent experiments are shown. **c** Quantitative RT-PCR analysis of markers after directed differentiation to definitive endoderm fate using FGF2 and activin A in WT or R255K mESCs. The expression of genes was first normalized to *Gapdh* then to that in undifferentiated mESCs. Data are mean ± s.d. from three independent experiments. Two-sided Student's *t*-test. **d** MeRIP-qPCR of m⁶A peaks on *Klf4*, *Smad6*, and *Smad7*. Enrichment of MeRIP versus input RNA was normalized against *Gapdh*. Data are mean ± s.d. from three independent experiments. Two-sided Student's *t*-test. **e** The stability of *Klf4*, *Smad6*, and *Smad7* RNAs in WT and R255K cells after Actinomycin D treatment. The expression level was normalized against *Gapdh* then to WT at 0 h. Data from three independent experiments are shown. Source data for (**b**–**e**) are provided as a Source Data file.

and uncover a specific role for it in mESC endoderm differentiation, suggesting a crosstalk between protein methylation and RNA methylation in gene expression.

## Methods

**Cell cultures**. E14Tg2a (National Collection of Authenticated Cell Cultures) and CGR8 (from Dr. Shoujun Huang) murine embryonic stem cells (mESCs) were cultured in high-glucose DMEM (Invitrogen) supplemented with 15% fetal bovine serum (FBS, Biological Industries), 1% non-essential amino acid (Invitrogen), 1 mM glutamine (Invitrogen), 0.15 mM 1-thioglycerol (Sigma-Aldrich), 100 U/mL Penicillin/Streptomycin (Invitrogen), and 1000 U/mL LIF (Millipore) at 37 °C with 5% CO₂. Cells were maintained in plates coated with 0.2% gelatin.

The human HeLa (from Dr. Qinmiao Sun) and HEK293T (National Collection of Authenticated Cell Cultures) cell lines were grown in DMEM media supplemented with 10% FBS (Biological Industries), 100 U/mL penicillin, and 100 U/mL streptomycin (Invitrogen), at 37 °C with an atmosphere of 5% CO2 and 95% humidity.

For EB differentiation, mESCs were cultivated in low-attachment dishes at a density of $1 \times 10^5$/mL in the presence of complete medium without LIF. EBs were collected at the indicated time points for further experiments.

Definitive endoderm differentiation of mESCs was carried out as described[57–59]. mESCs were hung at a concentration of 1000 cells per 20 µl drop in Dulbecco's Modified Eagle Medium containing Nutrient Mixture F-12 (DMEM/F12 1:1), 20% fetal bovine serum for stem cell, NEAA (Gibco), Glutamine (Gibco), and 100 µM 1-Thioglycerol (Sigma). 100 drops were plated per dish. After 4 days of growth, the cells were transferred to a gelatin-coated plate to adhere. 50 ng/ml activin A (R&D) and 5 ng/ml bFGF (R&D) were then used to induce the endoderm differentiation for 48 h.

**Stable cell line generation**. Coding regions of METTL14, METTL14 R255K, PRMT1, or GFP were cloned into pcDNA4-TRE-SFB vector[60] and transfected into HeLa cells with pEF-IRES-rtTA plasmids using PEI (Polyscience). 36 h later, 10 µg/mL puromycin (ThermoFisher) and 100 µg/mL zeocin (ThermoFisher) were added to the medium and selected for 24–48 h. Resistant cells were plated at low density, and single colonies were isolated. The protein expression was confirmed by western blot.

**siRNA KD of proteins**. siRNAs were synthesized by RiboBio (Guangzhou) and transfected into HeLa cells using Lipofectamine 2000 Transfection Reagent (ThermoFisher). 48 h later, cells were collected for RT-qPCR, WB or LC–MS/MS. The siRNA sequences are listed in supplementary table 1.

**Virus packaging and infection**. pLVX vectors containing GFP and PRMT1 CDS were transfected into 293 T cells with pIRES-rtTA, pMD2.G and psPAX2. Viruses were collected after 48–72 h, concentrated, and used to infect mESCs along with polybrene (8 mg/mL, Sigma). 48 h after infection, blasticidin and G418 were added to the culture medium to select for the infected cells. The surviving cells were collected for WB and LC–MS/MS.

**Generation of knock-in and knockout cell lines**. Knock-in cell lines were generated using CRISPR-Cas9 techniques[61,62]. sgRNAs recognizing genomic regions near the arginine 255 site were designed on http://crispr-era.stanford.edu/ and cloned into the pXPR_001 plasmid. Donor vectors were generated by integrating the PCR product of METTL14 genomic regions around R255 sites into a pUC19 vector, and the AGA at amino acid 255 was mutated to AAA to generate the R255K mutation (Supplementary Table 2). Donor and pXPR_001 were transfected by electroporation into mES cells using a Neo kit (Gibco). After 48 h, 2 µg/mL puromycin (ThermoFisher) was added, and resistent cells were plated for single colony isolation. Colonies with the desired mutation were identified by restriction enzyme digest and Sanger sequencing.

For knockout cell line generation, pXPR001-sgRNA (Supplementary Table 3) was transfected and selected as above. Colonies were identified by Sanger sequencing.

**Inhibitor treatment**. Adenosine, periodate oxidized (AdOx, Sigma) was dissolved in dimethyl sulfoxide (DMSO). A HeLa SFB-METTL14 stable cell line was treated with 30 µM AdOx and DMSO for 36 h before pulldown assays with S-protein agarose beads (Novagen). Arginine methylation was determined by western blot.

**Alkaline Phosphatase staining and quantification**. mESC colonies were fixed with 4% paraformaldehyde and processed using BCIP/NBT Alkaline Phosphatase Color Development Kit following the manufacturer's instructions (Beyotime). An Alkaline Phosphatase Assay Kit (Beyotime) was used to quantify ALP. A Microplate Reader was used to measure the OD at 405 nm.

**Cell proliferation assays**. For analysis of ESC self-renewal, mESCs (20,000 per well) were seeded in 12-well plates, and on days 1–5, cells were collected and counted. Alkaline phosphatase staining was performed with a BCIP/NBT Alkaline Phosphatase Color Development Kit (Beyotime) following the manufacturer's instructions.

**Tandem affinity purification (TAP) of S-Flag-SBP-tagged proteins**. For identification of METTL14 PTMs, HeLa cells stably expressing SFB-METTL14 were lysed in buffer (20 mM Tris-Cl at pH 7.4, 150 mM NaCl, 1 mM EDTA, 1 mM EGTA, 1% NP-40, 0.5% sodium deoxycholate, 0.1% SDS, and protease inhibitor (Sigma)) for 30 min at 4 °C. Crude lysates were treated for 40 sec with a sonicator (SCIENTZ, 10% power, 5 sec on/ 5 sec off), and the debris was removed. The cell lysates were incubated with streptavidin Sepharose beads (GE Healthcare), for 2 h at 4 °C, and bead-bound proteins were eluted with 1 mg/mL biotin (Sigma). Eluates were further subjected to S protein beads immunoprecipitation (Novagen), and bound proteins were subjected to SDS-PAGE and visualized by Coomassie blue staining (Tiangen). Protein bands were excised and subjected to LC–MS/MS analysis.

For the identification of METTL14 interacting proteins, the TAP procedure was performed as above, except for the buffer components (25 mM Tris-Cl pH 7.4, 150 mM NaCl, 0.5% TritonX-100, and 10% glycerol).

**LC–MS/MS analysis of protein**. The LC–MS/MS was carried out as described[63] (PTM biolab). Gel pieces were destained, dehydrated with 100% acetonitrile, and rehydrated by incubating with dithiothreitol. After a secondary dehydration, samples were incubated in 55 mM iodoacetamide for 45 min protected from light. Gel pieces were washed with NH₄HCO₃, dehydrated, and then digested with trypsin at 37 °C overnight.

Peptides were extracted, dried, and dissolved in solvent A (0.1% formic acid, 2% acetonitrile in water), and then loaded onto a reversed-phase analytical column packed in-house (15-cm length, 75 µm i.d.) and run on an EASY-nLC 1000 UPLC system at a constant flow rate of 800 nL/min. The gradient started from 6% to 25% solvent B (0.1% formic acid in 90% acetonitrile) in 16 min, then by a 6 min gradient from 25% to 40%, and finally climbing to 80% in 4 min, then holding at 80% for 4 min. The peptides were then subjected to nanospray ionization (NSI) (ThermoFisher) and analyzed by tandem mass spectrometry (MS/MS) using an Orbitrap Q Exactive Plus tandem mass spectrometer (ThermoFisher) coupled online to the UPLC. For the full MS scan, the range was set to 350–1800 m/z. Intact peptides were detected at a resolution of 70,000 in the Orbitrap. Peptides were selected for MS/MS using normalized collision energy (NCE) setting of 28. The ion fragments were detected at a resolution of 17,500 in the Orbitrap. A data-dependent procedure which alternated between one MS scan and 20 MS/MS scans was applied with 15.0 s dynamic exclusion. The electrospray voltage applied was 2.0 kV. Automatic gain control (AGC) was set at 5E4. The intensity threshold was set to 5E3, and the maximum injection time was 200 ms. The MS/MS data were processed using Proteome Discoverer 1.3. Tandem mass spectra were searched using Mascot search engines (v2.3.0) in Proteome Discoverer against METTL14 protein sequence database. Trypsin/P was set as the cleavage enzyme, and up to

two missing cleavages in phosphorylation and up to four missing cleavages in methylation detection were allowed. Mass error of precursor ions was set to 10 ppm and fragment ions were set to 0.02 Da. Carbamidomethyl on Cys was specified as a fixed modification, and phosphorylation on Ser, Thr, and Tyr, mono-methylation and di-methylation on Arg, oxidation on Met, and N-terminal acetylation were specified as variable modifications. To increase the confidence of identification, peptide confidence was set as high, and peptide ion score was set to >20.

For the identification of METTL14 interacting proteins, the tryptic peptides were dissolved in solvent A and loaded onto an Acclaim PepMap RSLC C18 column (15 cm length, 75 μm i. d.). The gradient was composed of an increase from 5% to 90% solvent B (0.1% formic acid in 80% acetonitrile) over 60 min at a constant flow rate of 300 nL/min on an LC-20AD (Shimadzu) and a Dionex Ultimate 3000 RSLCnano system (ThermoFisher). The peptides were then subjected to nanospray ionization (NSI) (ThermoFisher) and analyzed by tandem mass spectrometry (MS/MS) using an Orbitrap Q Exactive tandem mass spectrometer (ThermoFisher). Peptides were then selected for MS/MS using Normalized Collision Energy (NCE) setting of 27. Automatic gain control (AGC) was set at 1E5. The maximum IT for MS1 and MS2 were set to 40 ms and 60 ms. The MS/MS data were processed using Proteome Discoverer 1.3. Tandem mass spectra were searched using Mascot search engines (v2.3.0) in Proteome Discoverer. Mass error of precursor ions was set to 20 ppm, and fragment ions were set to 0.6 Da. Carbamidomethyl on Cys was specified as a fixed modification, and oxidation was specified as a variable modification. The significance threshold was set to 0.05.

**Western blots**. For NIR western blot, the secondary antibodies Dylight 800 Goat Anti-Rabbit IgG (1:10000, A23920, Abbkine), Dylight 800, and Goat Anti-Mouse IgG (1:10000, A23910, Abbkine) were used. Membranes were scanned using an Odyssey Clx Imager (LI-COR) and quantified using Image Studio 5.2 (LI-COR). The uncropped blots for all western blots were provided in the Source Data file.

**RNA pulldown assays**. Flag tagged METTL14 and METTL14 R255K transfected HeLa cells were collected using a precooled cell scraper and lysis buffer (10 mM Tris-HCl, pH 7.4, 1 mM EDTA, 0.2% Tween, 150 mM NaCl, protease inhibitor, and Rnase inhibitor (Promega)) for 30 min at 4 °C. Cells were briefly sonicated, and the debris was removed at 13,523 g for 10 min at 4 °C. RNA probes containing an $m^6A$ motif (5′biotin-GGACCRRACHGGUCCCCACGCGUCGRRACHCGA) (Takara) were incubated with MyOne Streptavidin C1 beads (ThermoFisher) for 15 min at room temperature, and redundant probes were abandoned. Cell supernatants were incubated with probe-beads complex for 2 h at 4 °C. Nonspecific binding was removed by washing with lysis buffer and 10 mM Tris-HCl, pH 7.4, 0.1% NP-40, and 150 mM NaCl. Pulldown proteins were eluted with 2x protein loading buffer for 10 min at 98 °C and detected by western blot.

**RT-qPCR**. Total RNA or immunoprecipitated RNA was reverse transcribed by a Promega GoScript Reverse Transcription System A500. The resulting cDNA was used for real-time quantitative polymerase chain reaction (qPCR) with a Roche LightCycler 96. Primers are listed in Supplementary Table 2. Data were analyzed using the $2^{-\Delta\Delta Ct}$ method.

**Cell fractionation and immunoblotting**. In total, $1 \times 10^7$ cells were collected and suspended in 800 μL 1 × cell lysis buffer (10 mM Tris pH 7.4, 10 mM NaCl, and 0.5% NP-40) freshly supplemented with protease inhibitor cocktail and PhosSTOP Tablets (Roche). Cells were incubated for 10 min at 4 °C. Nuclei were recovered after centrifugation at 376 g for 10 min at 4 °C. Supernatants had cytoplasmic components. Nuclei were washed three times with PBS. Nuclear and cytoplasmic proteins were denatured by 5X protein loading buffer (1 M Tris-Cl pH 6.8, 10% SDS, 50% glycerol, 10 mM DTT, and 0.5% bromophenol blue) for 10 min at 98 °C, and then subjected to western blot analysis. Primary antibodies used were as follows: mouse anti-WTAP (1:1000, Proteintech, 60188-1-Ig), rabbit anti-WTAP (1:1000, Abcam, ab195380 (EPR18744)), rabbit anti-METTL3 (1:3000, Bethyl, A301-567A), rabbit anti-METTL14 (1:1000, Sigma, HPA038002), rabbit anti-PRMT1 (1:1000, CST, 2449 (A33)), mouse anti-mono and dimethyl arginine (1:1000, Abcam, ab412), rabbit anti-mono-methyl arginine (1:1000, CST, 8015 S), rabbit anti-GAPDH (1:10000, Proteintech, 60004-1-Ig), mouse anti-flag (1:3000, MBL, M185-3L), mouse anti-MYC (1:1000, MBL, M192-3), and mouse anti-HA (1:3000, MBL, M180-3). Secondary antibodies used were as follows: anti-mouse (1:5000, KPL, 5220-0341), anti-rabbit (1:5000, KPL, 5220-0336), and VeriBlot for IP Detection Reagent (HRP) (1:200, Abcam, ab131366).

**Co-immunoprecipitation**. HeLa cells were collected by cell scraping and 211 g centrifugation for 2 min. Supernatants were discarded, and sediment was lysed in IP buffer (25 mM Tris, 150 mM NaCl, 1 mM EDTA, 5% glycerol, 1% Triton X-100, 1 mM PMSF, 1 mM DTT, protease inhibitor cocktail, and PhosSTOP Tablets) at 4 °C for 30 min to 1 h. Debris was removed at 18,407 g for 10 min, and supernatants were supplemented with equal amounts IP buffer without TritonX-100. Appropriate amounts of METTL14 (1:50, Sigma, HPA038002) antibody, PRMT1 (1:50, CST, 2449 (A33)) antibody, and control rabbit IgG (1:100, Santa, SC-2027) were incubated with supernatants and enriched with Dynabeads Protein G (A1)

(Invitrogen, 10004D) for 3 h at 4 °C. Nonspecific binding was removed by washing with IP buffer three times, and bead-bound proteins were eluted with 5× protein loading buffer for 10 min at 98 °C for SDS-PAGE for target protein detection. Co-IP in HeLa SFB-tagged METTL14/PRMT1 stable cell line was conducted using S-protein beads.

**In vitro arginine methylation assays**. In total, 0.5 μg peptides (R255 mono-methylated (RVCLRKWGYRR(Met)CED), R255 WT (RVCLRKWGYRRCED), and R255K mutation (RVCLRKWGYRKCED)) were incubated with 0 μg (−) or 0.1 μg (+) PRMT1 in 20μL reaction systems containing 50 mM Tris-HCl pH 8.6, 2 mM MgCl2, 1 mM TCEP (ThermoFisher, 20490), 0.02% Triton X-100, and 25 μM SAM for 3 h at room temperature. Dot blots were conducted in metal baths at 37 °C, and crosslinking was performed for 1 h at 37 °C. Arginine methylation signals of products were detected by WB using anti-mono-methyl arginine (CST, 8015 S) antibody.

**$m^6A$ dot blots**. Total RNA was extracted with TRNZOL (TIANGEN), and mRNA was purified twice with Dynabeads mRNA purification kits (ThermoFisher, cat#61006). Purified polyadenylated RNAs were denatured and spotted onto Amersham Hybond-N+ membranes (GE Healthcare) several times. After UV crosslinking at 2000 mJ/cm², unbound RNA was washed away with PBST buffer (0.05% Tween-20). After blocking with PBSTA buffer (0.05% Tween-20, 3% BSA) for 1 h, membranes were incubated with $m^6A$ antibody (1:3000, Synaptic Systems, 202003) overnight at 4 °C. After incubating with anti-rabbit IgG-HRP (1:5000, KPL, 5220-0336) in PBSTA (0.05% Tween-20, 3% BSA) for 2 h at room temperature, $m^6A$ signals were captured by Sage SmartChemi610. Membranes were immersed in methylene blue (MB) for normalization of spotted RNAs.

**Quantification of RNA $m^6A$ by LC–MS/MS**. Purified mRNA (200 ng–1 μg) was digested to nucleosides with 0.5 U nuclease P1 (Sigma, N8630) in 20 μL buffer containing 10 mM ammonium acetate, pH 5.3 for 6 h at 42 °C followed by addition of 2.5 μL 0.5 M MES buffer, pH 6.5, and 0.5 U CIAP (Takara, 2250 A) for 6 h at 37 °C. Mixtures were diluted to 60 μL, filtered and injected to An Agilent Poroshell 120 column coupled online to AB SCIEX Triple Quad 5500 LC mass spectrometer (Applied Biosystems) in positive electrospray ionization mode for LC–MS/MS analysis[64]. Concentrations of $m^6A$ were determined based on standard curves from pure nucleoside standards, and $m^6A/A$ ratios were calculated by concentration.

**Photoactivatable-ribonucleoside-enhanced crosslinking and immunoprecipitation qPCR**. Photoactivatable-ribonucleoside-enhanced crosslinking and immunoprecipitation was carried out as reported[10,65]. Briefly, mESCs stably transfected with SFB-METTL14 or SFB-METTL14 R255K were cultured with 100 μM 4SU (Aladdin) and 6 μg/ml doxycycline for 16 h. Cells were washed with PBS twice and crosslinked for 400 mJ/cm² under 365 nm UV light on ice. Cells were then collected, lysed with 50 mM Tris-HCl pH7.5, 100 mM NaCl, 2 mM EDTA, and 0.5% NP-40 supplemented with RNase Inhibitor and Protease Inhibitor Cocktail for 30 min and briefly sonicated. Then the lysates were centrifuged and the supernatants were collected. 1 U/μL RNaseT1 (ThermoFisher) was used to fragment RNA at room temperature for 8 min. In total, 20 μL flag beads (Genscript) was then used to bind SFB tagged proteins for 2 h at 4 °C. Beads were washed with high salt buffer once and low salt buffer twice. Finally, beads were incubated in 200 μl buffer containing 0.2 mg/ml Proteinase K at 56 °C for 15 min. The supernatant was collected, and the RNA was extracted by the TRNZOL Reagent (Tiangen). Then RT qPCR was performed.

**Molecular simulation**. To study the binding between METTL14 R255me and RNA substrates, molecular docking and molecular dynamics (MD) simulations were carried out (Modekeji). The initial structure of METTL14 was generated based on the reported crystal structure of METTL3-METTL14 complex (PDB ID 5IL1)[48]. To optimize the initial RNA structure, the RNA structure of GGACU was equilibrated in a box of water and ions and run to 20 ns for simulation[50]. RNA was docked into the RNA binding region of the METTL3-METTL14 complex[48] using HDOCK for semi-flexible docking[66]. The structure with top docking score (−210.90) was selected for further analysis.

MD simulation of the METTL14-RNA complex was carried out using YASARA[51,67]. The hydrogen-bonding network was optimized, and pKa prediction within YASARA based on the electrostatic potential, hydrogen bonds, and accessible surface area[68] was conducted to fine-tune the residues' protonation states at pH 7.4. 0.1538 mol/L NaCl ions were added. The unmethylated and methylated simulated system contains a total of 55599 and 55601 atoms, respectively. To remove clashes, steepest descent and simulated annealing minimizations were carried out. Simulations were run at a temperature of 298 K and a pressure of 1 atm (NPT ensemble), adopting TIP3P model for the water and AMBER14 force field for the protein and RNA[69], and the force field parameters of methylated R255 were automatically generated by the YASARA AutoSMILES algorithm[70–73]. The simulation of RNA-METTL14 was equilibrated for 10 nanoseconds (ns), during which coordinates of the backbone chain atoms of protein and RNA were restrained, and the amino acid side-chain atoms were sufficiently released. The simulation was then run for 50 ns for production. The cutoff of Van der Waals

forces was set to 1.0 nm. Particle mesh Ewald method was used for long-range electrostatics calculation with a cut-off of 1.0 nm[74]. Bonded interactions were set to 1.25 fs, and non-bonded interactions were set to 2.5 fs as described[67]. Root mean square deviation, radius of gyration, and solvent accessible surface area were adopted to ensure the equilibrium of the RNA-protein complex (Supplementary Fig. 6a–d). The root mean square fluctuation (RMSF) of RNA was measured over the 50 ns production time. The binding energy $E_{bind}$ was calculated by the YASARA software using the following equation:

$$E_{bind} = E_{pot\_complex} + E_{sol\_complex} - E_{pot\_METTL14} - E_{sol\_METTL14} - E_{pot\_RNA} - E_{sol\_RNA}$$

(1)

where $E_{pot\_complex}$, $E_{pot\_METTL14}$, and $E_{pot\_RNA}$ are the potential energy of the complex, METTL14, and RNA, respectively. The solvent energy component $E_{sol}$ was defined by:

$$E_{sol} = E_{solcoulomb} + E_{solvdw} + molsurf \times surfcost$$

(2)

where $E_{solcoulomb}$ and $E_{solvdw}$ are the coulomb and Van der Waals components of solvation energy, molsurf is molecular solvent accessible surface areas, and surfcost is a guesstimate of the entropic cost of exposing an area in the surface to the solvent in kJ/mol and 0.65 was utilized (http://www.yasara.org). Three simulation replicates were carried out.

**MeRIP-seq and qPCR.** MeRIP-seq was as conducted as described[75]. Briefly, total RNA was extracted and fragmented into ~100-nucleotide fragments with zinc acetate, and 300 μg fragmented RNA was prepared for immunoprecipitation with $m^6A$ antibodies (10 μg, Abcam, ab151230). Nonspecific binding was washed away with a stringent workflow: high-salt buffer (400 mM NaCl, 0.05% NP-40, 10 mM Tris-HCl), competitive buffer (150 mM NaCl, 0.05% NP-40, 10 mM Tris-HCl, 0.25 mg/mL adenosine, uridine, guanosine and cytidine mix), high-detergent buffer (150 mM NaCl, 0.5% NP-40, 10 mM Tris-HCl), and immunoprecipitation buffer (150 mM NaCl, 0.05% NP-40, 10 mM Tris-HCl). Bound $m^6A$ RNA was eluted by competition with 1 mg/mL $N^6$-methyladenosine (Selleckchem). For input RNA, ribosomal RNA was removed with Epicentre Ribo-zero rRNA removal kits (Epicentre). Library construction of immunoprecipitated RNAs and input RNAs was conducted using NEBNext Ultra RNA Library Prep Kits for Illumina (New England Biolabs) on an Illumina Hiseq Xten platform.

For MeRIP-qPCR, about 30 μg fragmented RNA and 1.5 μg $m^6A$ antibodies were utilized. Immunoprecipitated RNAs were reverse transcribed using a GoScript Reverse Transcription System (Promega, A500) and analyzed using real-time qPCR. Ratios of immunoprecipitated versus input for peaks were calculated and normalized to *GAPDH*. Primers are listed in Supplementary table 2.

**mRNA lifetime analysis.** mESCs were treated with 5 μg/ml Actinomycin D (JK chemical) for the indicated time and then harvested using TRNZOL. RNA was extracted and subjected to qPCR analysis or mRNA library construction using a NEBNext Ultra RNA Library Prep Kit (NEB). Sequencing was carried out on the Illumina Hiseq Nova platform.

**RNA-seq analysis.** Reads that mapped to rRNA fasta sequences from the UCSC gene annotation (mm10) using bowtie2 (v.2.2.6)[76] were discarded, and the remaining reads were aligned to mm10 using hisat2-align-s (v.2.0.5)[77]. Unique reads with high mapping quality were retained using Picard (http://broadinstitute.github.io/picard, v.2.16.0) and SAMtools (v.0.1.09)[78]. Differential gene expression between WT and METTL14 R255K was analyzed by DESeq2 using default parameters[79].

For mRNA lifetime RNA-seq data analysis, FPKM of genes were calculated, and the lifetime was calculated as reported[8].

**Peak and differential peak identification.** Regions of significant MeRIP enrichment relative to input control ("peaks") were identified using MACS2[80]. The parameter '--nomodel' was used for macs2 peak calling. Macs2 bdgdiff was used to identify differential peaks. The enrichment of biological process was conducted using Kobas[81].

**Statistical analysis and reproducibility.** The number of replicated times of western blots, micrographs, LC–MS/MS or qPCR experiments is indicated in the figure legends. Key experiments were repeated in another cell line or another knockout or knock-in clone. Triplicates were performed for key experiments. Similar results were obtained in each case, and a representative figure is shown. The statistical comparisons between two groups were performed using an unpaired two-tailed Student's $t$-test or Mann–Whitney $U$-test. Graphpad Prism software (8.3.0) was used for Student's $t$-test analysis. The exact $P$ values, $t$ value, and degrees of freedom were provided in the source data file. *$P < 0.05$; **$P < 0.01$; ***$P < 0.001$; ****$P < 0.0001$; NS not significant.

**Reporting Summary.** Further information on research design is available in the Nature Research Reporting Summary linked to this article.

## Data availability
MeRIP–seq data has been deposited in GEO under accession numbers GSE164047. The mass spectrometry data has been deposited in integrated proteome resources (iProX) under accession number IPX0002127000 and ProteomeXchange Consortium under ID PXD018458. The data supporting the findings of this study are available from the corresponding authors upon reasonable request. Source data are provided with this paper.

## Code availability
Custom code written in Python is available on Github (https://github.com/linjian-smu/METTL14_R255K.git) and Zenodo[82] (https://doi.org/10.5281/zenodo.4761865).

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

## Acknowledgements

This work was supported by the National Key R&D Program of China (2019YFA0802300, 2018YFC1004103 and 2017YFA0106700), the Natural Science Foundation of China (31590830, 81771643, 31970595, 31900588 and 31988101), Science and Technology Program of Guangzhou, China (202002030495).

## Author contributions

X.L. conceived the research; S.X., D.C., and L.X. designed and supervised the project; K.T., Z.W., J.J., H.W., Q.W., X.Z., E.D., and Q.L performed the experiments; Linjian.X. and J. C. conducted the bioinformatics analysis; W. D. conducted the simulation; X.L., S. X., and L.X. wrote the manuscript with input from all authors.

## Competing interests

The authors declare no competing interests.
