## [Peer Review File · Nature Communications]

REVIEWER COMMENTS

Reviewer #1 (Remarks to the Author):

In this manuscript, Liu et al. reported that an arginine (R255) methylation in METTL14 is important for RNA m6A modification and endoderm differentiation of mouse embryonic stem cells (mESCs). The R255 methylation of METTL14 is mediated by PRMT1, and regulates the interaction between METTL14 and WTAP, further affecting the binding capacity of the m6A methyltransferase complex (MTC) with substrate mRNA. They also showed the effect of R255 methylation on mESC lineage commitment through targeted mRNA methylation. Overall, this is an interesting, innovative and timely study, which reveals the functional importance of arginine methylation of a key component of the m6A MTC on RNA modification and gene regulation. Nevertheless, to further improve the quality of this work, a number of concerns need to be addressed by the authors.

Major concerns:

1. To improve the rigor of this study, key experiments, such as those shown in Figures 1a, 3b-c, 4a-b, and 5a-d, should be repeated in at least one more cell line.
2. Figure 1a showed the R255 methylation of epitope labeled METTL14 overexpressed in HeLa cells. What percentage (me/total) of the peptides detected are methylated at R255 according to the mass spec analysis?
3. In Figures S2a and S2b, can the authors show HinP1I digestion sites in the sequence, or at least in Figure S2a scheme? Furthermore, the sgRNA sequences shown in Figures S2a and S2b seem to be different. Please double check. There are many synonymous mutations were generated in the target sites. There should be multiple clones generated from a successful CRISPR experiment, why did the authors choose this clone?
4. In Figures 2a and 2b, the expression level of METTL14 should be shown here, as the synonymous mutations might cause a differential level of expression.
5. In Figure 3F, how many amino acids were included in the peptides? In the R255K mutant, around half of the arginine methylation was left. The authors explained that there are other methylation sites on METTL14. If this is the case, are there any other methylation sites detected by the mass spec?
6. In Figure 5d, the level of endogenous WTAP, rather than the epitope labeled one, should be detected upon pulled down by METTL14, as the endogenous level of WTAP is high.
7. It would also be important to determine whether PRMT1 overexpression and knockdown affect the binding of METTL14 with METTL3 and WTAP in cells.
8. In Fig. 6e, the authors claimed in the manuscript that "It was well known that the m6A could promote the RNA degradation^{8, 9}". Actually, Reference 9 (i.e., Huang et al. Nature Cell Biology. 2018) reported that IGF2BP family proteins can promote the stability and translation of target transcripts as a new class of m6A readers. Thus, this sentence should be changed to "It was well known that the m6A modification could promote or inhibit target RNA degradation^{8, 9}". The RNA stability level changes of a few candidate targets shown in Figure 6e seem not significant. Can the authors show their RNA expression level changes? In addition, are there any target genes (especially those that have been reported to be important for mESC self-renewal or differentiation) whose RNA levels are decreased in the R255K cells than in the WT cell? If yes, please also check their RNA stability changes.

Minor concerns:

9. Figure S2C (left panel), the label 'Exon' was incomplete.
10. Fig. 5d, the positions of the labels did not match with the columns in the panel.
11. In Page 9, line 178, "did not changed" should be changed to "did not change"; line 180, "functional redundancy" should be changed to "functional compensation".

Reviewer #2 (Remarks to the Author):

RNA m6A methylation is important for a variety of biological processes including stem cell self-renewal

and differentiation and cancer progression. Modulation of m6A writer complex METTL3-METTL14-WTAP is likely cause cell type specific changes in m6A profiles. In this manuscript, Liu et al suggest that methylation of a specific Arg (R255) of METTL14 by PRMT1 is important for efficient m6A modification of mRNAs and endoderm differentiation of mouse embryonic stem cells. Three major points are made by authors: 1. METTL14 is methylated by PRMT1 at R255; 2. The R255 methylation of METTL14 promotes the interaction between METTL14 and WTAP therefore the stabilization of methylation enzyme complex, and the binding of METTL14 with RNA substrates; 3. The R255 methylation of METTL14 is required for endoderm differentiation of ESCs. While the study is of interest to both RNA modification and stem cell field, each point still need more supporting evidence. My major concerns:

1. Is PRMT1 the key enzyme modifying METTL14 in HeLa or mouse ESCs? This question can be answered by showing the level of R255 methylation of METTL14 in PRMT1 knockdown HeLa and PRMT1 knockout ESCs.
2. Is R255 methylation of METTL14 and/or PRMT1 developmentally regulated?
3. Simulations to show how R255 methylation affects the binding between METTL14 and RNA substrates should be performed.
4. MeRIP-Seq revealed both m6A downregulated and non-downregulated mRNAs. The binding between METTL14 and RNA substrates should be tested for both categories (RIP-PCR and/or RNA pull down).
5. To make definite conclusion on the impact of R255 methylation on endoderm differentiation, directed differentiation into endoderm lineages should be performed.
6. The mechanistic explanation of endoderm differentiation defects is very superficial. Functional rescue experiments are preferred (e.g. knocking down key genes in R255K cells).
7. The quality of the study need be improved: A) At least two clones from knockout or knockin should be checked, alternatively, key findings should be verified in another clone; B) Preferentially, rescue experiments should be done for PRMT1 knock out as well R255K knock in cells; C) Triplicates should be performed for key experiments, quantifications with appropriate statistics analysis should be conducted. (e.g. Fig1c, 3e, 4d, 5d, 5e, 5f, 6d, 6e and more) ; D) The marker size should be shown along with all western blots.
8. Previous studies have suggested that decrease in m6A methylation promotes the self-renewal of naive ESCs. However, this study observed the loss of AP activity and downregulation of pluripotency genes. Can the authors comment on this discrepancy? Can authors prove that their ESCs are still naïve ESCs (high nanog, esrrb, nanog, rex1 expression, low otx2, fgf5 et al)?

Minor points:

1. Figure 1d, high resolution structure with RNA substrates should be shown.
2. Why R255 regulate lower abundance m6A, high expression genes? What are sequence features on RNA substrates that leads to R255 regulation?
3. Line 179, "Therefore, the increase of WTAP protein in the R255K cells may due to functional redundancy caused by the decreased interaction between WTAP with METTL14-METTL3."; "functional redundancy" does not make sense here. I guess the authors meant secondary effects?
4. What about the expression of Klf4 and Smad6/7 in R255K ESCs as well as during EB differentiation?
5. Do PRMT1 knockout have similar differentiation phenotype as R255K knockin?
6. Figure 5d, the labelling of samples are off to the right.
7. Global analysis of mRNA half-life in R255K ESCs should be done if possible. This will likely give more insights on the functional target of R255 methylated METTL14.
8. Did authors check sequences around R255K knock in position? How far? This might be important if authors want to make sure R255K but not other unwanted mutations cause the phenotype.
9. Figure S5b, high resolution pictures are preferred. One cannot make definite conclusion based on these data. Alternatively, the authors may quantify the fraction of differentiated, partially differentiated and undifferentiated ESC colonies.

Reviewer #3 (Remarks to the Author):

In the manuscript by Shan Xiao and colleagues, the authors are describing a new post-translation modification of METTL14 (a methylation of Arg 255) and its potential contribution on RNA methylation of Adenosine (m6A) and its role in the endoderm differentiation of embryonic stem cells.

The data shown in the manuscript suggests that PRMT1 is responsible for the methylation of METLL14 R255 (metR255), which in turn stabilises the methyltransferase complex METTL14-MTTL3-WPAT onto its substrate to promote the methylation of mRNA m6A.

Different experimental techniques have been used to support/probe the proposed mechanism of actions. LC-MS/MS combined with affinity purification (pull downs) of overexpressed METTL14 and PRMT1, have been used to identify PTMs on METTL14 and binding partners. Nevertheless, there is no much information on how the LC-MS/MS and the data analysis was carried on. This is very important to be able to judge the quality and integrity of the MS data. On those lines, below are the points that the authors should be addressing regardless the mass spectrometry side of the manuscript.

Specific comments related to LC-MS/MS data:

It was noted that the mass spectrometry data is not uploaded in any Mass spectrometry repositories such as Pride (proteome Xchange), could the authors make the data accessible to the community?

MATERIALS AND METHODS SECTION

1. Page31, from line 544: Tandem affinity purification (TAP) of S-Flag-SBP-tagged proteins
There is a quite comprehensive description on how MTTL14 was pulled down and eluted from the beads. However, there is no reference or detailed information on how the in-gel-digest, the LC-MS/MS and data analysis were performed. The only thing provided is the link to the company that did the LC-MS/MS analysis in which is not easy to find the technical details. For that reason, I would ask the authors to expand in the following matters and/or provide references:

1.1 In-gel-digest: Please provide a brief description of the protocol used specifying what protease they did use. It seems that the in-gel digest approach was used both for the PTM detection in METTL14 (Fig 1-page18 and suppl. Fig1-page 39) and the identification partners of METTL14 (fig 3; page 21).

1.2 LC-MS/MS: Brief description of the LC-MS/MS set up used including the type of mass spectrometer used (brand), LC gradient, column types, type mass analysers used for MS1 and MS2 scans etc.

1.3 Data analysis: The only information provided is the software used (PD vs 1.3). Please indicate the basic parameters search (i.e. data search engine, database used, proteolytic enzyme, missed cleavages, modifications set up, mass accuracy windows). Could you also indicate what filters were used (if any) to increase confidence of the identification?

RESULTS

2. Page 5, from line 85, section "METTL14 is methylated at arginine 255"

2.1 The authors show that they are able to express SFB-tag-METTL14 at similar endogenous level as shown in fig 1a, upon Dox treatment. The WB shows some levels of inducible SFB-tag-METTL14 in absence of Dox. Could the authors explain why and clarify?

2.2 Detection of METTL14 methylation at R255 by LC-MS/MS.

The WB (fig 1a) confirms that METTL14 is methylated. To narrow down the position, SFB-METTL14 was pull down and subjected to in-gel digest and LC-MS/MS. The identification of the Rme255 is based only on one PSM (Suppl. Fig 1c). The MS/MS spectra of the methylated peptide (Fig 1b) seems to indicate that the methylation could be in the R255, judging mainly from the b ions detected (b4 to b8), however the authors do not show the mass error of the peptide identification. Here, there are some points that should be addressed:

2.3 Suppl. Fig 1b (page 39), shows a Coomassie blue stained gel with a faint band identified as METTL14 and METTL3 (MW difference of ~12kDa). How much was loaded into the gel? The image does not display the MW of each marker lane. Could the authors please add the MW reference at least in the marker below and above the band of interest?

2.4 Suppl. Fig 1c (page 39) includes a table with the modified peptides identified and matched to METTL14 and METTL3 with some of the identification parameters (sequence coverage, scores, site of the modification etc.). Could you also add here the mass error for each peptide? In line to my comments regarding material and methods (point1), knowing the search engine used it will help to interpret how the score parameters are determined. Why or what's the relevance of showing the PTMs on METTL3 when PTMS on METTL3 are not described or discussed in the manuscript?

2.5 The sequence coverage (%) shown per each peptide is a bit confusing. How is it calculated? Is it expressed related to each peptide or to the protein? Please clarify this.

2.6 Fig 1b (page 18) MS/MS spectra of METTL14 peptide containing Rme255: Could the authors add the mass of each b ion?

2.7 Confirmation of the Rme255 by mutating SFB-METTL14 255aa from R to a K (line 92; Fig1c page 18): WB against MMA shows a decrease of mono-methylation when R255 is mutated to a K, which sort of validated the Mass spec data. However, the WB band is still fairly intense. How do the authors explain this? The data provided does not show the detection of any other methylation site on METTL14.

3 Fig 3 (page 21) and Suppl. Fig 3a (page 43), Regarding METTL14 interacting with PRMT1: In line with my comments in point 1 about expanding the data analysis workflow used for the LC-MS/MS data, could you please indicate in fig 3a legend, what data search engine were used for protein identification?

Regarding suppl. Fig 3a, I understand this sample was processed using the same in-gel digest and LC-MS/MS workflow as used for detecting PTM on METTL14?

4 Fig 5 d (page 25): WB is not aligned with the legend displaying each lane treatment.

Reviewers: Benedikt M Kessler and Iolanda Vendrell

Response to Reviewers

Responses to Comments by Reviewer#1

General comments: “*In this manuscript, Liu et al. reported that an arginine (R255) methylation in METTL14 is important for RNA m⁶A modification and endoderm differentiation of mouse embryonic stem cells (mESCs). The R255 methylation of METTL14 is mediated by PRMT1, and regulates the interaction between METTL14 and WTAP, further affecting the binding capacity of the m⁶A methyltransferase complex (MTC) with substrate mRNA. They also showed the effect of R255 methylation on mESC lineage commitment through targeted mRNA methylation. Overall, this is an interesting, innovative and timely study, which reveals the functional importance of arginine methylation of a key component of the m⁶A MTC on RNA modification and gene regulation. Nevertheless, to further improve the quality of this work, a number of concerns need to be addressed by the authors.*”

Response: We are very grateful for the positive appraisal on our manuscript that “***this is an interesting, innovative and timely study, which reveals the functional importance of arginine methylation of a key component of the m⁶A MTC on RNA modification and gene regulation***”. As suggested, we have conducted a number of additional experiments to strengthen our paper.

Major concerns:

1. *To improve the rigor of this study, key experiments, such as those shown in Figures 1a, 3b-c, 4a-b, and 5a-d, should be repeated in at least one more cell line.*

Response 1: We thank the reviewer for this constructive suggestion. Accordingly, we've repeated these experiments in another cell line, and consistent results were obtained. The SFB-METTL14 protein in E14Tg2a mouse embryonic stem cells was also arginine methylated and the methylation level was remarkably decreased upon treatment with the arginine methylation inhibitor AdOx (revised Supplementary Fig. 1b). The interaction of endogenous METTL14 and PRMT1 was further confirmed in HEK293T cells (revised Supplementary Fig. 3b-c). The mRNA m⁶A level change in HeLa cells in response to PRMT1 over-expression or knock down (shown in Fig. 4a-b) was repeated in PRMT1 over-expression and *Prmt1* knockout E14Tg2a cells (now shown in revised Fig. 4e and 4c, respectively). We've generated another METTL14 R255K knock-in cell in CGR8 mouse embryonic stem cells, and found that the protein level of METTL3 and METTL14 were unchanged while WTAP was increased in R255K cells (revised Supplementary Fig. 5a). The cellular localization of METTL14/METTL3/WTAP were also unchanged (revised Supplementary Fig. 5a). We also checked the interaction of METTL14 with METTL3 and WTAP in HEK293T cells, and found that the interaction with METTL3 was not changed, while that with WTAP was remarkably decreased in R255K (revised Supplementary Fig. 5b).

2. *Figure 1a showed the R255 methylation of epitope labeled METTL14 overexpressed in Hela cells. What percentage (me/total) of the peptides detected are methylated at R255 according to the mass spec analysis?*

Response 2: We thank the reviewer for this comment. To get the intensity of peptides, we analyzed the MS/MS data using the Maxquant search engine (v.1.5.2.8) against the sequence of METTL14 concatenated with reverse decoy database (Le-tian, et al. *BMC Genomics* 2020: 435) (PTM biolab). Then the intensity of R255 methylated peptides and unmethylated peptides were used for the ratio calculation, giving an estimated ratio of methylated peptide/ total peptide of about 36.6%.

3. In Figures S2a and S2b, can the authors show HinP1I digestion sites in the sequence, or at least in Figure S2a scheme? Furthermore, the sgRNA sequences shown in Figures S2a and S2b seem to be different. Please double check. There are many synonymous mutations were generated in the target sites. There should be multiple clones generated from a successful CRISPR experiment, why did the authors choose this clone?

Response 3: We appreciate this comment from the reviewer. We apologize for the inconsistency of sgRNA sequence in the figures, which we have now corrected in the revised Supplementary Fig. 2a. The HinP1I digestion site is now shown in the scheme (revised Supplementary Fig. 2a). The synonymous mutations in knock-in cells were introduced to facilitate screening using restriction endonuclease digestion and to avoid re-cutting by changing the sgRNA targeted sequence (Yang, et al. *Nature Protocols* 2014: 1956-1968, Yao, et al. *Cell Research* 2017: 801-814). Multiple clones with the same mutations were generated, and we randomly picked a clone to perform the experiments.

4. In Figures 2a and 2b, the expression level of METTL14 should be shown here, as the synonymous mutations might cause a differential level of expression.

Response 4: We thank the reviewer for this suggestion. The expression level of METTL14 was not changed in R255K cells as shown in original Fig. 5a, and we've also presented another panel of METTL14 levels in WT and R255K in revised Fig. 2a following this suggestion.

5. In Figure 3F, how many amino acids were included in the peptides? In the R255K mutant, around half of the arginine methylation was left. The authors explained that there are other methylation sites on METTL14. If this is the case, are there any other methylation sites detected by the mass spec?

Response 5: We thank the reviewer for this comment. Fourteen amino acids were included in the peptides, and the sequence of the peptide is RVCLRKKGWYRRCD. The remaining mono-methylated signal may be due to the other arginine sites in the peptides that could also be methylated by PRMT1 *in vitro*. On our hand, R255 is the only methylation site detected by mass spec in the peptide. Since the discovery rate for methylated peptides are relatively low using mass spec (Pang, et al. *Bmc Genomics* 2010: 92-92, Larsen, et al. *Science Signaling* 2016: rs9-rs9, Musiani, et al. *Science Signaling* 2019: eaat8388, Ong, et al. *Nature Methods* 2004: 119-126), there might be other methylated sites on METTL14.

6. In Figure 5d, the level of endogenous WTAP, rather than the epitope labeled one,

should be detected upon pulled down by METTL14, as the endogenous level of WTAP is high.

Response 6: We thank the reviewer for this constructive suggestion. Accordingly, we've carried out the co-IP of METTL14 with the endogenous WTAP, which showed that the interaction between METTL14 R255K and endogenous WTAP was also decreased (see revised Supplementary Fig. 5c).

7. It would also be important to determine whether PRMT1 overexpression and knockdown affect the binding of METTL14 with METTL3 and WTAP in cells.

Response 7: We thank the reviewer for this constructive suggestion and carried out co-IP experiments to test this. We found that after PRMT1 knockdown, the binding between METTL14 and WTAP was weakened, while the binding between METTL14 and METTL3 was not substantially changed (revised Supplementary Fig. 5i), which is consistent with the results in R255K (revised Fig. 5d, Supplementary Fig. 5b-c). The changes of the binding between METTL14 and METTL3/WTAP in PRMT1 overexpression cells were not detected (revised Supplementary Fig. 5j).

8. In Fig. 6e, the authors claimed in the manuscript that “It was well known that the m⁶A could promote the RNA degradation^{8, 9}”. Actually, Reference 9 (i.e., Huang et al. Nature Cell Biology. 2018) reported that IGF2BP family proteins can promote the stability and translation of target transcripts as a new class of m⁶A readers. Thus, this sentence should be changed to “It was well known that the m⁶A modification could promote or inhibit target RNA degradation^{8, 9}”. The RNA stability level changes of a few candidate targets shown in Figure 6e seem not significant. Can the authors show their RNA expression level changes? In addition, are there any target genes (especially those that have been reported to be important for mESC self-renewal or differentiation) whose RNA levels are decreased in the R255K cells than in the WT cell? If yes, please also check their RNA stability changes.

Response 8: We thank the reviewer for this insightful comment. We've changed the sentence to “Given that m⁶A can promote or inhibit targeted RNA degradation (Wang, et al. Nature 2014: 117-20, Huang, et al. Nat Cell Biol 2018: 285-295)...” following the suggestion.

To obtain a whole picture of R255 target genes which are important for mESC self-renewal or differentiation, we first measured the RNA expression atlas in WT and R255K mESCs using RNA-seq. We found that the expression level of most genes with R255 regulated m⁶A peaks were not substantially changed (revised Supplementary Fig. 6f), including *Klf4*, *Smad6* and *Smad7*. For those genes that were differentially expressed, we found that they were not enriched in self-renewal or differentiation related processes (revised Supplementary Table 4). Then we measured the mRNA stability using RNA-seq after transcription inhibition with Actinomycin-D in WT and R255K mESCs. We found that over 67.8% of R255 targeted genes degraded slower in R255K than that in WT (fold change of half-life > 1.5, revised Supplementary Fig. 6g), including *Klf4*, *Smad6*, *Smad7* and some other pluripotency genes and differentiation genes. By contrast, there were about 5% of transcripts that degraded faster in R255

mESCs than in WT, and these are probably targets of IGF2BP.

Integrating the RNA expression and stability data, we found that although the genes all have decreased m⁶A in R255K, most of the genes have inconsistent stability and steady state RNA level. This phenomenon is very intriguing, suggesting that there might be other mechanisms regulating the RNA level, something we plan to explore in our future work.

Minor concerns:

9. Figure S2C (left panel), the label 'Exon' was incomplete.

Response 9: We apologize for this mistake and have corrected it in the revised Supplementary Fig. 2d.

10. Fig. 5d, the positions of the labels did not match with the columns in the panel.

Response 10: We apologize for this mistake and have corrected it in the revised Fig. 5d.

11. In Page 9, line 178, "did not changed" should be changed to "did not change"; line 180, "functional redundancy" should be changed to "functional compensation".

Response 11: We thank the reviewer for this suggestion. "did not changed" was changed to "did not change". The sentence about "functional redundancy" was changed to "This suggests the increase of WTAP protein in the R255K cells may be an attempt to compensate for the decreased interaction between WTAP with METTL14-METTL3".

Responses to Comments by Reviewer #2

General comments: "RNA m⁶A methylation is important for a variety of biological processes including stem cell self-renewal and differentiation and cancer progression. Modulation of m⁶A writer complex METTL3-METTL14-WTAP is likely cause cell type specific changes in m⁶A profiles. In this manuscript, Liu et al suggest that methylation of a specific Arg (R255) of METTL14 by PRMT1 is important for efficient m⁶A modification of mRNAs and endoderm differentiation of mouse embryonic stem cells. Three major points are made by authors: 1. METTL14 is methylated by PRMT1 at R255; 2. The R255 methylation of METTL14 promotes the interaction between METTL14 and WTAP therefore the stabilization of methylation enzyme complex, and the binding of METTL14 with RNA substrates; 3. The R255 methylation of METTL14 is required for endoderm differentiation of ESCs. While the study is of interest to both RNA modification and stem cell field, each point still need more supporting evidence."

Response: We are very grateful for the positive appraisal on our manuscript that "**the study is of interest to both RNA modification and stem cell field**". As suggested, we have provided extensive additional evidences to strengthen our points.

Major concerns:

1. Is PRMT1 the key enzyme modifying METTL14 in HeLa or mouse ESCs? This question can be answered by showing the level of R255 methylation of METTL14 in

PRMT1 knockdown HeLa and PRMT1 knockout ESCs.

Response 1: We thank the reviewer for this constructive suggestion. For lack of the antibody specific for R255 methylation, we attempted to address this issue using a pan mono-methyl arginine antibody. To reduce the interference of other potential methylation sites, we generated a METTL14 truncation (aa 147-395, 409-452, METTL14-T) and found the mono-methyl arginine level was greatly reduced when R255 is mutated (R255K-T) (revised Supplementary Fig. 3f), suggesting that R255 is one of the main methylation sites in this truncation. This is in agreement with our previous LC-MS/MS results for arginine methylation (revised Fig. 1b, revised Supplementary Fig. 1d). Importantly, the mono-methylation of METTL14-T, but not R255K-T, was reduced in response to PRMT1 knockdown (revised Supplementary Fig. 3f), indicating that PRMT1 does modify METTL14 at R255 in HeLa cells. We sought to more definitively demonstrate the role of PRMT1 as suggested, transfecting METTL14-T into *Prmt1*^{-/-} mESCs, but unfortunately, we failed to detect the expression of proteins. Nevertheless, additional evidence further supports our conclusion: we have previously demonstrated that PRMT1 physically interacts with METTL14 and efficiently methylates R255 *in vitro* (revised Fig. 3a-f and Supplementary Fig. 3b-e). In fact, *Prmt1* is the predominant type I arginine methyltransferase and is responsible for 85% of arginine methyltransferase activity in mammals (Tang, et al. *J Biol Chem* 2000: 7723-30, Pawlak, et al. *Mol Cell Biol* 2000: 4859-69). Together, these results demonstrate that PRMT1 is responsible for R255 methylation, although we can't completely exclude the participation of other PRMTs.

2. Is R255 methylation of METTL14 and/or PRMT1 developmentally regulated?

Response 2: We thank the reviewer for this excellent comment. We also hope to thoroughly study the dynamics of R255 methylation *in vivo*, particularly in a developmental context, but to do this requires an effective antibody against endogenous METTL14 R255 mono-methylation. Generating antibodies specifically recognizing protein methylation has been challenging (Hattori and Koide *Current opinion in structural biology* 2018: 141-148, Hattori, et al. *Nature methods* 2013: 992-995), and although we have put a lot of effort into making such an antibody, we have not developed an effective one. Nevertheless, we note that the m⁶A level assessed by LC-MS/MS is decreased in EBs relative to ESCs (Response Fig. 1a). The expression level of *Prmt1* also decreased, while the expression of *Mettl3* and *Mettl14* were not substantially changed (Response Fig. 1b) (Geula, et al. *Science* 2015: 1002-1006). Consistent with this, we found that the m⁶A peaks decreased in EBs were enriched in R255 methylation regulated ones (Fisher's exact test, $P < 1.91 \times 10^{-9}$, odds ratio = 1.755). Thus, it could be possible that R255 is developmentally regulated in the process of ESC differentiation.

3. Simulations to show how R255 methylation affects the binding between METTL14 and RNA substrates should be performed.

Response 3: We thank the reviewer for this helpful comment. Following the suggestion, we investigated the binding of RNA with R255 unmethylated/methylated METTL14

using molecular docking and molecular dynamics simulations (see method “molecular simulation” section). Briefly, we generated the initial structure of METTL14 based on the crystal structure of the METTL3-METTL14 complex (PDB ID 5IL1) (Wang, et al. *Nature* 2016: 575-+) and introduced a methylation group at R255. The RNA structure of GGACU was equilibrated in a box of water and ions, and run to 20 ns for the simulation using YASARA (Krieger, et al. *Methods Mol Biol* 2012: 405-21). Then RNA was docked onto the RNA binding region of METTL3-METTL14 complex using HDOCK (Yan, et al. *Nucleic Acids Research* 2017: W365-W373). Root mean square deviation, radius of gyration and solvent accessible surface area were adopted to ensure the equilibrium of RNA-protein complex. The root mean square fluctuation (RMSF) of RNA in R255 methylated complex was lower than that in R255 unmethylated complex, suggesting that the structure of RNA in R255 methylated complex is more stable (Supplementary Fig.5d).

The molecular dynamics (MD) of RNA with R255 unmethylated or mono-methylated METTL14 was simulated using YASARA (Krieger, et al. *Methods Mol Biol* 2012: 405-21, Krieger and Vriend *Bioinformatics* 2014: 2981-2982). We measured the hydrogen bond interaction and binding energy of RNA with METTL3-METTL14, and found that after R255 mono-methylation, the average number of hydrogen bonds increased from 5.14 to 7.09 (revised supplementary Fig.5e), and the binding energy was decreased from -214.000 ± 134.86 kJ/mol to -657.660 ± 98.94 kJ/mol after 30 ns simulation (revised supplementary Fig.5f), indicating the binding of RNA with METTL3-METTL14 was more stable after R255 mono-methylation. Moreover, we found that R255 methylation changed the conformation of the sidechain, causing the change of RNA binding pattern on protein surface (revised supplementary Fig.5g). The 5' phosphate group of RNA turned to R471 from His401 after R255 was methylated (revised supplementary Fig.5h).

Together, we found that the binding of RNA with METTL3-METTL14 was more stable after R255 mono-methylation using molecular simulation, which is consistent with the previous findings showing R255 methylation enhanced the RNA binding ability of METTL14 (revised Fig.5e-f).

4. MeRIP-Seq revealed both m⁶A downregulated and non-downregulated mRNAs. The binding between METTL14 and RNA substrates should be tested for both categories (RIP-PCR and/or RNA pull down).

Response 4: We thank the reviewer for this suggestion, and accordingly have shown the binding between METTL14 and RNA substrates of both categories in revised Fig 5e. Consistently, we found that the METTL14 R255K's RNA-binding ability for transcripts with decreased m⁶A modification in R255K cells is significantly reduced, and the binding ability for transcripts with non-downregulated m⁶A is not substantially changed.

5. To make definite conclusion on the impact of R255 methylation on endoderm differentiation, directed differentiation into endoderm lineages should be performed.

Response 5: We thank the reviewer for this constructive suggestion. We directed the

mESCs to differentiate to definitive endoderm (DE) following the suggestion using activating A and bFGF (Shirasawa, et al. *Biochem Biophys Res Commun* 2011: 608-13, Chen, et al. *Development (Cambridge, England)* 2013: 675-686, Wang and Yang *J Vis Exp* 2008). We found that while WT mESCs successfully generated the markers of DE, R255K mESCs failed to differentiate to endoderm lineages (revised Fig. 6c), suggesting that R255 methylation is required for normal mESC endoderm differentiation.

6. *The mechanistic explanation of endoderm differentiation defects is very superficial. Functional rescue experiments are preferred (e.g. knocking down key genes in R255K cells).*

Response 6: We thank the reviewer for this constructive suggestion. To give more insight into the functional target of R255 methylated METTL14, we measured the global mRNA stability in WT and R255K mESCs and found many pluripotency genes and differentiation genes degraded slower in R255K than in WT (revised Supplementary Fig. 6g). Among these genes are *Smad6/7*, known antagonists of activin-Nodal-TGF β signaling (Pauklin and Vallier *Development* 2015: 607-19), and *Klf4*, an endoderm differentiation inhibitor (Aksoy, et al. *Nat Commun* 2014: 3719). We then knocked down *Klf4* and *Smad6/7* in R255K cells and found the expression of endoderm genes was partially rescued in EB differentiation (revised Supplementary Fig. 6j-k).

7. *The quality of the study need be improved: A) At least two clones from knockout or knockin should be checked, alternatively, key findings should be verified in another clone; B) Preferentially, rescue experiments should be done for PRMT1 knock out as well R255K knock in cells; C) Triplicates should be performed for key experiments, quantifications with appropriate statistics analysis should be conducted. (e.g. Fig 1c, 3e, 4d, 5d, 5e, 5f, 6d, 6e and more) ; D) The marker size should be shown along with all western blots.*

Response 7: We thank the reviewer for these constructive suggestions.

- A) We've checked another clone of METTL14 R255K and *Prmt1*^{-/-} and verified the mRNA m⁶A level decrease (shown in revised Supplementary Fig. 2c and 4e). We've also generated another R255K knock-in clone in CGR8 cells and verified the expression level and subcellular location of METTL14 and other methyltransferase components were not substantially changed (revised Supplementary Fig. 5a).
- B) Rescue experiments were carried out in PRMT1 knockout and R255K knock-in cells, and the m⁶A levels were partially restored to WT levels in these experiments (revised Fig. 2c and Fig. 4c).
- C) Western blots (Fig. 1c, 3e, 5d) were performed at least three times, and the replicates are shown in response Fig.2. At least triplicates and statistical analyses for Fig. 4d, 5e, 6d, 6e and other key experiments (Fig. 2c, 2f) are provide in revised Fig. 4d, 5e, 6d, 6e, 2c and 2f, as the reviewer suggested. The repetitive times and statistical analyses used for all experiment are included in the figure legends, and the reproducibility is described in the "Statistics and reproducibility" section.

D) The marker sizes have been added to western blots as the reviewer's suggested.

8. *Previous studies have suggested that decrease in m6A methylation promotes the self-renewal of naive ESCs. However, this study observed the loss of AP activity and downregulation of pluripotency genes. Can the authors comment on this discrepancy? Can authors prove that their ESCs are still naive ESCs (high nanog, esrrb, nanog, rex1 expression, low otx2, fgf5 et al)?*

Response 8: We thank the reviewer for raising this interesting point. As reported (Geula, et al. *Science* 2015: 1002-1006), the mESCs cultured in 2i conditions tend to remain in a ground naïve state, while the primed mESCs tend to differentiate, and metastable mESCs cultured in FBS demonstrate an intermediate response regarding Mettl3 depletion. And the down-regulated pluripotency of R255K mESCs expanded in FBS in our study is consistent with previous reports (Wang, et al. *Nature Cell Biology* 2014: 191-198) showing that mESCs cultured in FBS conditions lose self-renewal capability upon m⁶A loss after knockdown of Mettl3 or Mettl14. Accordingly, we've checked the expression of relevant genes and confirmed that the R255K cells are still pluripotent with high *Nanog*, *Esrrb*, *Rex1*, *Sox2* and low *Otx2*, *Fgf5*, *Gata6*, *Gata4* expression (revised Supplementary Fig. 6d).

Minor concerns:

1. *Figure 1d, high resolution structure with RNA substrates should be shown.*

Response 1: We thank the reviewer for this suggestion and have shown the high resolution structure of METTL14/METTL3 with RNA substrates in revised Fig. 1d.

2. *Why R255 regulate lower abundance m6A, high expression genes? What are sequence features on RNA substrates that leads to R255 regulation?*

Response 2: We thank the reviewer for this comment. The R255 regulated m⁶A regions are enriched for the motif A(G/A)AC, while the constitutive m⁶As are enriched for the motif GGAC (revised Supplementary Fig. 2f). According to the structural basis reported (Wang, et al. *Nature* 2016: 575-+), A(G/A)AC motif may not be an ideal substrate for METTL14, thus the R255 dependent m⁶A peaks may have lower m⁶A abundance. And as previously reported (Meyer, et al. *Cell* 2012: 1635-46), lower abundance m⁶A peaks are often observed in highly expressed transcripts. Consistent with this, the expression levels of genes associated with these R255 regulated m⁶A regions were significantly higher than the other genes in WT cells.

3. *Line 179, "Therefore, the increase of WTAP protein in the R255K cells may due to functional redundancy caused by the decreased interaction between WTAP with METTL14-METTL3."; "funcational redundancy" does not make sense here. I guess the authors meant secondary effects?*

Response 3: We thank the reviewer for this suggestion and changed the sentence to "This suggests the increase of WTAP protein in the R255K cells may be an attempt to compensate for the decreased interaction between WTAP with METTL14-METTL3" accordingly.

4. *What about the expression of Klf4 and Smad6/7 in R255K ESCs as well as during EB differentiation?*

Response 4: We thank the reviewer for this comment. The expression level of *Klf4* and *Smad6/7* in R255K mESCs were not significantly changed compared with that in WT mESCs (Supplementary fig. 6f and 6i). During EB differentiation, *Smad6* in WT cells was slightly increased, while the other ones were all decreased (Supplementary Fig.6i).

5. *Do PRMT1 knockout have similar differentiation phenotype as R255K knockin?*

Response 5: We thank the reviewer for this comment. Our PRMT1 knockout mESCs were unable to differentiate into rounded spheres, consistent with the previous reports showing that PRMT1 mutation leads to early embryonic lethality (Pawlak, et al. *Mol Cell Biol* 2000: 4859-69, Yu, et al. *Molecular and Cellular Biology* 2009: 2982-2996). Since Prmt1 is the predominant type I arginine methyltransferase (Tang, et al. *J Biol Chem* 2000: 7723-30, Pawlak, et al. *Mol Cell Biol* 2000: 4859-69), we speculated that other targets of Prmt1 along with METTL14 R255 might be responsible for the differentiation failure of *Prmt1*^{-/-} mESCs.

6. *Figure 5d, the labelling of samples are off to the right.*

Response 6: We apologize for this mistake and have corrected it in the revised Fig. 5d.

7. *Global analysis of mRNA half-life in R255K ESCs should be done if possible. This will likely give more insights on the functional target of R255 methylated METTL14.*

Response 7: We thank the reviewer for this constructive suggestion. As suggested, we measured the mRNA stability using RNA-seq after transcription inhibition with Actinomycin-D in WT and R255K mESCs. We found that over 67.8% of R255 targeted genes degraded slower in R255K than in WT (fold change of half-life > 1.5, revised Supplementary Fig. 6g), including *Klf4*, *Smad6*, *Smad7* and some other pluripotency genes and differentiation genes.

8. *Did authors check sequences around R255K knock in position? How far? This might be important if authors want to make sure R255K but not other unwanted mutations cause the phenotype.*

Response 8: We thank the reviewer for this suggestion. We have checked the sequence 143 nt upstream and 226 nt downstream of the R255K knock-in position using Sanger sequencing, and no other mutations were detected. We further checked the RNA sequencing data of METTL14 R255K and confirmed that there are no other mutations on the exon of METTL14. We've also confirmed some findings in another clone, and R255K knock-in in another cell line (CGR8) to rule out the effect of unwanted mutations.

9. *Figure S5b, high resolution pictures are preferred. One cannot make definite conclusion based on these data. Alternatively, the authors may quantify the fraction of differentiated, partially differentiated and undifferentiated ESC colonies.*

Response 9: We thank the reviewer for this suggestion. Accordingly, high resolution pictures and quantification of AP staining are presented in revised Supplementary Fig. 6b, showing that the R255K cells displayed less alkaline phosphatase (AP) positive colonies.

Responses to Comments by Reviewer #3

General comments: “In the manuscript by Shan Xiao and colleagues, the authors are describing a new post-translation modification of METTL14 (a methylation of Arg 255) and its potential contribution on RNA methylation of Adenosine (m6A) and its role in the endoderm differentiation of embryonic stem cells.”

The data shown in the manuscript suggests that PRMT1 is responsible for the methylation of METLL14 R255 (metR255), which in turn stabilises the methyltransferase complex METTL14-MTTL3-WPAT onto its substrate to promote the methylation of mRNA m6A.

Different experimental techniques have been used to support/probe the proposed mechanism of actions. LC-MS/MS combined with affinity purification (pull downs) of overexpressed METTL14 and PRMT1, have been used to identify PTMs on METTL14 and binding partners. Nevertheless, there is no much information on how the LC-MS/MS and the data analysis was carried on. This is very important to be able to judge the quality and integrity of the MS data. On those lines, below are the points that the authors should be addressing regardless the mass spectrometry side of the manuscript.

Response: We thank the reviewer for the constructive suggestions and revised the manuscript accordingly.

Specific comments related to LC-MS/MS data:

It was noted that the mass spectrometry data is not uploaded in any Mass spectrometry repositories such as Pride (proteome Xchange), could the authors make the data accessible to the community?

Response: We thank the reviewer for this suggestion. We have uploaded the mass spectrometry data on integrated proteome resources (iProX) under accession number IPX0002127000, ProteomeXchange ID PXD018458. The reviewer’s URL is <https://www.iprox.org/page/PSV023.html?url=1608086968639fnTs>, and the password is ROd3.

MATERIALS AND METHODS SECTION

1. Page31, from line 544: Tandem affinity purification (TAP) of S-Flag-SBP-tagged proteins

There is a quite comprehensive description on how MTTL14 was pulled down and eluted from the beads. However, there is no reference or detailed information on how the in-gel-digest, the LC-MS/MS and data analysis were performed. The only thing provided is the link to the company that did the LC-MS/MS analysis in which is not easy to find the technical details. For that reason, I would ask the authors to expand in the following matters and/or provide references:

1.1 In-gel-digest: Please provide a brief description of the protocol used specifying

what protease they did use. It seems that the in-gel digest approach was used both for the PTM detection in METTL14 (Fig 1-page18 and suppl. Fig1-page 39) and the identification partners of METTL14 (fig 3; page 21).

Response: We thank the reviewer for this constructive suggestion and have included the information in the methods section (see Method “LC-MS/MS analysis of protein” section). In-gel digestion was used to detect the PTMs and partners of METTL14 (Li, et al. *Nature Immunology* 2016: 806-815). Briefly, gel pieces were destained, dehydrated with 100% acetonitrile and rehydrated by incubating with dithiothreitol. After a secondary dehydration, samples were incubated in 55 mM iodoacetamide for 45 min at dark. Gel pieces were washed with NH_4HCO_3 , dehydrated, and then digested with trypsin at 37 °C overnight. Peptides were then extracted, dried and subjected to LC-MS/MS analysis.

1.2 LC-MS/MS: Brief description of the LC-MS/MS set up used including the type of mass spectrometer used (brand), LC gradient, column types, type mass analysers used for MS1 and MS2 scans etc.

Response: We thank the reviewer for this suggestion. The LC-MS/MS was carried out essentially as aforementioned (Li, et al. *Nature Immunology* 2016: 806-815). The tryptic peptides were dissolved in solvent A (0.1% formic acid, 2% acetonitrile in water), and then loaded onto a home-made reversed-phase analytical column (15-cm length, 75 μm i.d.). The gradient was comprised of an increase from 6% to 25% solvent B (0.1% formic acid in 90% acetonitrile) over 16 min, 25% to 40% in 6 min and climbing to 80% in 4 min then holding at 80% for the last 4 min, all at a constant flow rate of 800 nL/min on an EASY-nLC 1000 UPLC system. The peptides were subjected to NSI source followed by tandem mass spectrometry (MS/MS) in Q Exactive (Thermo Fisher Scientific) coupled online to the UPLC. The electrospray voltage applied was 2.0 kV. The scan range was set to 350 to 1800 m/z for full MS scan, and intact peptides were detected in the Orbitrap at a resolution of 70,000. Peptides were then selected for MS/MS using normalized collision energy (NCE) setting as 28 and the fragments were detected in the Orbitrap at a resolution of 17,500. A data-dependent procedure that alternated between one MS scan followed by 20 MS/MS scans with 15.0s dynamic exclusion was used. Automatic gain control (AGC) was set at 5E4, with an intensity threshold of 5E3 and a maximum injection time of 200 ms.

1.3 Data analysis: The only information provided is the software used (PD vs 1.3). Please indicate the basic parameters search (i.e. data search engine, database used, proteolytic enzyme, missed cleavages, modifications set up, mass accuracy windows). Could you also indicate what filters were used (if any) to increase confidence of the identification?

Response: The MS/MS data were processed using Proteome Discoverer 1.3. Tandem mass spectra were searched using Mascot search engines against METTL14 protein sequence database. Trypsin/P was set as the cleavage enzyme, and up to 2 missing cleavages in phosphorylation and up to 4 missing cleavages in methylation detection were allowed. Mass error of precursor ions was set to 10 ppm and fragment ions was

set to 0.02 Da. Carbamidomethyl on Cys was specified as a fixed modification, and phosphorylation on Ser, Thr and Tyr, mono-methylation and di-methylation on Arg, oxidation on Met and N-terminal acetylation were specified as variable modifications. To increase the confidence of identification, peptide confidence was set as high, and peptide ion score was set to > 20.

RESULTS

2. Page 5, from line 85, section “METTL14 is methylated at arginine 255”.

2.1 The authors show that they are able to express SFB-tag-METTL14 at similar endogenous level as shown in fig 1a, upon Dox treatment. The WB shows some levels of inducible SFB-tag-METTL14 in absence of Dox. Could the authors explain why and clarify?

Response: We thank the reviewer for this helpful comment. The tet-on transcriptional activation system has low background expression in the absence of Dox induction (Loew, et al. *Bmc Biotechnology* 2010: 81-81). We re-ran the samples adopting Near-Infrared (NIR) western blot, to quantify the background expression of METTL14. It's about 2.5% of induced expression (revised Supplementary Fig. 1a). We think this background expression is too weak to interfere with the results.

2.2 Detection of METTL14 methylation at R255 by LC-MS/MS.

The WB (fig 1a) confirms that METTL14 is methylated. To narrow down the position, SFB-METTL14 was pull down and subjected to in-gel digest and LC-MS/MS. The identification of the Rme255 is based only on one PSM (Supp. Fig 1c). The MS/MS spectra of the methylated peptide (Fig 1b) seems to indicate that the methylation could be in the R255, judging mainly from the b ions detected (b4 to b8), however the authors do not show the mass error of the peptide identification. Here, there are some points that should be addressed:

Response: We thank the reviewer for this constructive suggestion. Mass error of the peptide identification was added in revised Supplementary Fig. 1d.

2.3 Suppl. Fig 1b (page 39), shows a Coomassie blue stained gel with a faint band identified as METTL14 and METTL3 (MW difference of ~12kDa). How much was loaded into the gel? The image does not display the MW of each marker lane. Could the authors please add the MW reference at least in the marker below and above the band of interest?

Response: We thank the reviewer for this constructive suggestion. Eighteen 10 cm dishes of cells were subjected to purification of SFB-METTL14. The cells were lysed, sonicated and METTL14 in lysate was bound by streptavidin Sepharose beads, washed and competitive eluted using biotin. Then the eluates were further subjected to an S protein beads immunoprecipitation. All of the purified proteins were loaded into the gel. Judging from the BSA titration on the left lanes, less than 1 µg of SFB-METTL14 was obtained. Molecular weights of the markers have been added to the image as suggested.

2.4 Suppl. Fig 1c (page 39) includes a table with the modified peptides identified and

matched to METTL14 and METTL3 with some of the identification parameters (sequence coverage, scores, site of the modification etc.). Could you also add here the mass error for each peptide? In line to my comments regarding material and methods (point1), knowing the search engine used it will help to interpret how the score parameters are determined.

Why or what's the relevance of showing the PTMs on METTL3 when PTMS on METTL3 are not described or discussed in the manuscript?

Response: We thank the reviewer for this constructive suggestion. We have added the mass error for each peptide in the revised Supplementary Fig 1d. The detailed information of LC-MS/MS was provided in the response to 'MATERIALS AND METHODS SECTION'. The search engine is Mascot. The PTMs on METTL3 is irrelevant to this current work, and we have removed them in the revised paper.

2.5 The sequence coverage (%) shown per each peptide is a bit confusing. How is it calculated? Is it expressed related to each peptide or to the protein? Please clarify this.

Response: We thank the reviewer for this comment. The sequence coverage (%) was calculated by dividing the number of amino acid sequences identified in LC-MS/MS by the number of amino acids in the METTL14 sequence.

2.6 Fig 1b (page 18) MS/MS spectra of METTL14 peptide containing Rme255: Could the authors add the mass of each b ion?

Response: We thank the reviewer for this suggestion and have added the mass of each b ion in revised Fig. 1b.

2.7 Confirmation of the Rme255 by mutating SFB-METTL14 255aa from R to a K (line 92; Fig1c page 18): WB against MMA shows a decrease of mono-methylation when R255 is mutated to a K, which sort of validated the Mass spec data. However, the WB band is still fairly intense. How do the authors explain this? The data provided does not show the detection of any other methylation site on METTL14.

Response: We thank the reviewer for this insightful comment. The discovery rate for methylated peptides are relatively low using mass spec, probably due to the substoichiometric nature of methylation (Pang, et al. *Bmc Genomics* 2010: 92-92, Larsen, et al. *Science Signaling* 2016: rs9-rs9, Musiani, et al. *Science Signaling* 2019: eaat8388), or inefficient trypsin cleavage at C-terminal to the methylated arginine and lysine residues (Ong, et al. *Nature Methods* 2004: 119-126). Though we only detected R255 methylation using mass spec in this study, there might be other methylated sites on METTL14.

3 Fig 3 (page 21) and Suppl. Fig 3a (page 43), Regarding MTTLE14 interacting with PRMT1:

In line with my comments in point 1 about expanding the data analysis workflow used for the LC-MS/MS data, could you please indicate in fig 3a legend, what data search engine were used for protein identification?

Regarding suppl. Fig 3a, I understand this sample was processed using the same in-gel

digest and LC-MS/MS workflow as used for detecting PTM on METTL14?

Response: We thank the reviewer for this comment. The same in-gel digestion was used to detect the partners of METTL14 in Fig.3 and Supplementary Fig.3a. Then the tryptic peptides were dissolved in solvent A (0.1% formic acid, 2% acetonitrile in water) and loaded onto an Acclaim PepMap RSLC C18 column (15 cm length, 75 μ m i. d.). The gradient was comprised of an increase from 5% to 90% solvent B (0.1% formic acid in 80% acetonitrile) over 60 min at a constant flow rate of 300 nL/min on an LC-20AD (Shimadzu) and Dionex Ultimate 3000 RSLCnano system (Thermo). The peptides were subjected to NSI source followed by tandem mass spectrometry (MS/MS) in Thermo Scientific Q Exactive. The scan range was 350 to 1800 m/z for full MS scan, and intact peptides were detected in the Orbitrap at a resolution of 70,000. Peptides were then selected for MS/MS using NCE setting at 27, and the fragments were detected in the Orbitrap at a resolution of 17,500. Automatic gain control (AGC) was set at 1E5. The maximum IT for MS1 and MS2 were set to 40 ms and 60 ms.

The MS/MS data were processed using Proteome Discoverer 1.3. Tandem mass spectra were searched using Mascot search engines. And we have added the information of search engine in the figure legend of Fig.3a accordingly. Trypsin/P was set as the cleavage enzyme, and up to 2 missing cleavages were allowed. Mass error of precursor ions was set to 20 ppm, and fragment ions was set to 0.6 Da. Carbamidomethyl on Cys was specified as a fixed modification, and oxidation was specified as a variable modification. To increase the confidence of identification, the significance threshold was set to 0.05.

4 Fig 5 d (page 25): WB is not aligned with the legend displaying each lane treatment.

Response: We apologize for this mistake and have corrected it in the revised Fig 5d.

Reference

1. Le-tian, Z. *et al.* Protein acetylation in mitochondria plays critical functions in the pathogenesis of fatty liver disease. *BMC Genomics* **21**, 435 (2020).
2. Yang, H., Wang, H.Y. & Jaenisch, R. Generating genetically modified mice using CRISPR/Cas-mediated genome engineering. *Nature Protocols* **9**, 1956-1968 (2014).
3. Yao, X. *et al.* Homology-mediated end joining-based targeted integration using CRISPR/Cas9. *Cell Research* **27**, 801-814 (2017).
4. Pang, C.N.I., Gasteiger, E. & Wilkins, M.R. Identification of arginine- and lysine-methylation in the proteome of *Saccharomyces cerevisiae* and its functional implications. *Bmc Genomics* **11**, 92-92 (2010).
5. Larsen, S.C. *et al.* Proteome-wide analysis of arginine monomethylation reveals widespread occurrence in human cells. *Science Signaling* **9**, rs9-rs9 (2016).
6. Musiani, D. *et al.* Proteomics profiling of arginine methylation defines PRMT5 substrate specificity. *Science Signaling* **12**, eaat8388 (2019).
7. Ong, S.E., Mittler, G. & Mann, M. Identifying and quantifying in vivo methylation sites by heavy methyl SILAC. *Nature Methods* **1**, 119-126 (2004).
8. Wang, X. *et al.* N6-methyladenosine-dependent regulation of messenger RNA stability. *Nature* **505**, 117-20 (2014).

9. Huang, H. *et al.* Recognition of RNA N(6)-methyladenosine by IGF2BP proteins enhances mRNA stability and translation. *Nat Cell Biol* **20**, 285-295 (2018).
10. Tang, J. *et al.* PRMT1 is the predominant type I protein arginine methyltransferase in mammalian cells. *J Biol Chem* **275**, 7723-30 (2000).
11. Pawlak, M.R., Scherer, C.A., Chen, J., Roshon, M.J. & Ruley, H.E. Arginine N-methyltransferase 1 is required for early postimplantation mouse development, but cells deficient in the enzyme are viable. *Mol Cell Biol* **20**, 4859-69 (2000).
12. Hattori, T. & Koide, S. Next-generation antibodies for post-translational modifications. *Current opinion in structural biology* **51**, 141-148 (2018).
13. Hattori, T. *et al.* Recombinant antibodies to histone post-translational modifications. *Nature methods* **10**, 992-995 (2013).
14. Geula, S. *et al.* m6A mRNA methylation facilitates resolution of naïve pluripotency toward differentiation. *Science* **347**, 1002-1006 (2015).
15. Wang, X. *et al.* Structural basis of N-6-adenosine methylation by the METTL3-METTL14 complex. *Nature* **534**, 575-+ (2016).
16. Krieger, E., Dunbrack, R.L., Jr., Hoof, R.W. & Krieger, B. Assignment of protonation states in proteins and ligands: combining pKa prediction with hydrogen bonding network optimization. *Methods Mol Biol* **819**, 405-21 (2012).
17. Yan, Y., Zhang, D., Zhou, P., Li, B. & Huang, S.-Y. HDOCK: a web server for protein–protein and protein–DNA/RNA docking based on a hybrid strategy. *Nucleic Acids Research* **45**, W365-W373 (2017).
18. Krieger, E. & Vriend, G. YASARA View—molecular graphics for all devices—from smartphones to workstations. *Bioinformatics* **30**, 2981-2982 (2014).
19. Shirasawa, S. *et al.* Pancreatic exocrine enzyme-producing cell differentiation via embryoid bodies from human embryonic stem cells. *Biochem Biophys Res Commun* **410**, 608-13 (2011).
20. Chen, A.E., Borowiak, M., Sherwood, R.I., Kweudjeu, A. & Melton, D.A. Functional evaluation of ES cell-derived endodermal populations reveals differences between Nodal and Activin A-guided differentiation. *Development (Cambridge, England)* **140**, 675-686 (2013).
21. Wang, X. & Yang, P. In vitro differentiation of mouse embryonic stem (mES) cells using the hanging drop method. *J Vis Exp* (2008).
22. Pauklin, S. & Vallier, L. Activin/Nodal signalling in stem cells. *Development* **142**, 607-19 (2015).
23. Aksoy, I. *et al.* Klf4 and Klf5 differentially inhibit mesoderm and endoderm differentiation in embryonic stem cells. *Nat Commun* **5**, 3719 (2014).
24. Wang, Y. *et al.* N6-methyladenosine modification destabilizes developmental regulators in embryonic stem cells. *Nature Cell Biology* **16**, 191-198 (2014).
25. Meyer, K.D. *et al.* Comprehensive analysis of mRNA methylation reveals enrichment in 3' UTRs and near stop codons. *Cell* **149**, 1635-46 (2012).
26. Yu, Z., Chen, T., Hébert, J.e., Li, E. & Richard, S.p. A Mouse PRMT1 Null Allele Defines an Essential Role for Arginine Methylation in Genome Maintenance and Cell Proliferation. *Molecular and Cellular Biology* **29**, 2982-2996 (2009).
27. Li, X. *et al.* Methyltransferase Dnmt3a upregulates HDAC9 to deacetylate the kinase TBK1 for activation of antiviral innate immunity. *Nature Immunology* **17**, 806-815 (2016).
28. Loew, R., Heinz, N., Hampf, M., Bujard, H. & Gossen, M. Improved Tet-responsive promoters

with minimized background expression. *Bmc Biotechnology* **10**, 81-81 (2010).

Response Figures

Response Fig. 1

Response Fig. 1

- LC-MS/MS quantification of m⁶A abundance in mRNA from ESCs and EBs. Data are mean \pm s.d. of four independent experiments.
- The expression level of *Prmt1*, *Mettl14* and *Mettl3* in ES and EB. Three input RNA-seq data were used to calculate the expression level (Geula, et al. *Science* 2015: 1002-1006).

Response Fig. 2

a

b

c

Response Fig. 2

- Replicates for Fig. 1c.
- Replicates for Fig. 3e.
- Replicates for Fig. 5f.

REVIEWER COMMENTS

Reviewer #1 (Remarks to the Author):

In this revised manuscript, the authors have added additional data, and have carefully and thoroughly addressed all the critiques from the reviewers. As a result, the quality of this paper has been further improved. Overall, this is an interesting and timely study. Therefore, I fully support the publication of this paper in Nature Communications.

Reviewer #2 (Remarks to the Author):

The manuscript has been significantly improved and most of my questions and comments are addressed. A few minor issues:

1. Figure S5i, WTAP (also METTL3) protein level appears to be decreased upon PRMT1 knockdown ; Can authors give a quantification for these protein level normalized to GAPDH? If WTAP is decreased, which is in sharp contrast to the increase of WTAP in R255K mutant, can authors comment on this?
2. Figure S3f, it is better to show quantification of two replicates, MMA signal versus Flag.
3. PRMT1^{-/-} show much higher m6A/A ratio than R255K mutant ESCs, which means either PRMT1 is not the major methyltransferase for R255 or R255K mutation has other impact besides blocking methylation. Can authors comment on these two possibilities? In addition, despite significant differences of m6A/A ratio between PRMT1^{-/-} and R255K mutant (3 fold), MeRIP enrichment showed similar decreases (Figure 2f and 4d) for PRMT1^{-/-} and R255K mutant, can authors explain this discrepancy?
4. Figure 5e, please label genes in each category (m6A decreased, m6A nondownregulated).
5. Figure 6j-k, the authors have tried very hard on the mechanistic explanation for defects in endoderm differentiation. However, these experiments are not conclusive and poorly carried out. First, genes selected for knocking down (Smad7 and Klf4) are not even upregulated in R255K mutant; Second, Only Klf4 is knocked down. I suggest the authors removing this part to avoid confusing future readers.
6. For newly added rescue experiments (Figure 2c and 4c), rescue level for PRMT1 and Mettl14 should be compared to the level of these genes in WT ESCs, preferably by Western blotting.
7. Figure 2a, quantification for three replicates should be added. Are these technical replicates or independent experiments?

Reviewer #3 (Remarks to the Author):

Upon receipt of the reviewer's criticisms, the authors have thoroughly revised their manuscript including new data as well as additional details into the methodology sections. This provides improved confidence for the characterisation of this posttranslational modification in METTL14.

Although the essential information is now provided, including public data repository, some of the newly added sections need further editing to be grammatically correct. Below are a couple of examples:

Minor comments

Line 675: "Mascot search engines" Please provide details, such as version, in-house or public etc.

Line 684: For the identification of METTL14 interacting proteins....

Line 688: ..and a Dionex Ultimate..

Lines 689/699: "The peptides were subjected to NSI source" - what does this mean? Please clarify. Something along the lines of:

The peptides were subjected to electrospray ionisation ESI and analysis by tandem mass spectrometry (MS/MS) using an Orbitrap Q Exactive tandem mass spectrometer (ThermoFisher).

Line 691: Please specify NCE - collision energy

Line 694: Mascot search engines - please specify as described above.

Lines 862/863: The mass spectrometry data has been deposited in integrated proteome resources, such as and

Reviewer #4 (Remarks to the Author):

I have been specifically asked to concentrate on the modeling and simulation aspects of the manuscript. My expertise lies far from the experimental biochemistry which constitutes the bulk of this work so I will abstain from commenting on that. The simulations were explicitly requested by one of the reviewers and are specifically aimed at clarifying the effect of R255 methylation of METTL14 on the binding of the METTL3-METTL14 complex to RNA substrates. The authors address this issue by performing molecular dynamics simulations of the methylated and unmethylated METTL3-METTL14 complex bound to a GGACU RNA ligand. Results are presented basically in the form of three observables followed in the course of the two simulations (methylated and unmethylated): 1) RMSF of the RNA nucleotides; 2) Number of protein-RNA hydrogen bonds; 3) Binding energy (unspecified, see below) between protein and RNA. The raw data used to generate the plots is also supplied in the form of a separate spreadsheet file, I found this quite helpful.

The idea of measuring the difference in stability of the methylated vs unmethylated systems from equilibrium simulations is quite reasonable. Unfortunately, the data presented are simply insufficient to draw meaningful conclusions (50ns total simulation time per system). This type of analysis generally requires much more sampling of conformational space, with several independent simulation runs. Ideally, to fully resolve such a question, a full-fledged binding free energy (not just enthalpy) study would be necessary. But I understand that this would constitute a new research project on its own and definitely would lie beyond the scope of the present study. Here, it would suffice to run a few more replicas of these simulations, to check that the results are not just a statistical accident. This should not be too problematic to run with YASARA on consumer grade hardware.

The main conclusion of greater stability of the methylated complex derived from the simulations rests on results presented in lines 205-209 of the manuscript: 1) the increase in the number of hydrogen bonds and concomitant decrease in the binding energy and 2) the lower RMSF values of the RNA nucleotides observed upon methylation of R255. The first are only seen in the last 20ns of the simulations, this is really too short a time and the relative difference is too small to exclude that these are not just an effect of statistical fluctuations. For the lower RMSF values it is not clear whether these are calculated over the entire 50ns or simply the last 20ns as above (see minor comments below). In either case, while the trend does point to a greater stability of the methylated complex, this could also be an effect of a bad or not well equilibrated starting structure. Only additional replicated simulations

can help clarify this.

Additionally, I have a some concerns regarding the simulations presented here:

Major:

The methods section is inadequate. For some of the following issues a bit of clarification is sufficient, but in order to allow others to reproduce/understand what was done, a lot more information needs to be specified:

- What is the final size of the simulated system in number of atoms including water, salt, and counter-ions?
- Why was the RNA ligand first equilibrated before docking (Lines 798-801)? Was HDOC used only for rigid body docking?
- What options were chosen in HDOC and what final scores were obtained? How were the docked structures selected? Is the large difference in the RNA structures shown in Fig S5g the result of the docking procedure, the H-bond optimization step, or the MD simulations? It makes a big difference if the starting points for the MD simulations are very similar and then diverge during the simulation, or if the simulations simply started from very different RNA conformations.
- The simulation protocol is poorly described, what forcefield parameters were used for the methylated R255? What was the equilibration protocol (were any restraints used, for how long)? The authors need to clearly state how long was the full equilibration time vs the production time.
- What were the criteria for deciding that the system had reached equilibrium? The authors mention RMSD, Rg, and SAS but do not specify how they chose 30ns to start production rather than some other time point. The RMSD, Rg and SAS data should also be supplied as separate plots (my suggestion would be to aggregate all simulation and modeling data in a separate figure).
- What pKa prediction algorithm was used to determine protonation states and with what parameters?
- How was the binding energy computed? Is this just derived from the intermolecular forcefield terms? Does this adequately represent the binding enthalpy? What about entropic contributions to binding?

Minor:

Line 194: "simulated" not "stimulated"

Line 199-204: Was RMSF calculated only during the equilibration phase and not during the "production phase of the simulations"? This seems unlikely to me, but the sentence starting at line 201 "We next simulated..." seems to imply simulation was carried out after calculation of RMSF? Please clarify the order of events here.

Use of units should be precise and consistent. The salt concentration used in the simulations is described in Line 806 as 0.9% NaCl. Is this molar percentage, mass percentage, volume percentage...? My suggestion is to use molar concentrations as one would for any experimental buffer solution. Distances: cutoffs are expressed in Å on Line 810, and in nm on Line 811, please choose one of the two.

In Fig S5g the choice of colors makes it very hard to distinguish the RNA from the protein background, would it be possible to change the color scheme (maybe also highlight the methylation point) to facilitate the reader?

Line 796: Who or what is Modekeji?

Response to Reviewers

Response to Comments by Reviewer #1

In this revised manuscript, the authors have added additional data, and have carefully and thoroughly addressed all the critiques from the reviewers. As a result, the quality of this paper has been further improved. Overall, this is an interesting and timely study. Therefore, I fully support the publication of this paper in Nature Communications.

Response: We are grateful for the positive appraisal of our manuscript that “*this is an interesting and timely study*”.

Response to Reviewer #2

The manuscript has been significantly improved and most of my questions and comments are addressed. A few minor issues:

Response: We thank the reviewer for the comments and suggestions, and accordingly, we have added additional data and comments to address the minor issues raised.

1. Figure S5i, WTAP (also METTL3) protein level appears to be decreased upon PRMT1 knockdown ; Can authors give a quantification for these protein level normalized to GAPDH? If WTAP is decreased, which is in sharp contrast to the increase of WTAP in R255K mutant, can authors comment on this?

Response: We thank the reviewer for this helpful comment. Following the suggestion, we carefully repeated *PRMT1* knockdown and western blot experiments for multiple times, and then quantified the WTAP and METTL3 level in all the replicates. We did not find a substantial decrease of WTAP and METTL3 in *PRMT1* knockdown cells (response Fig.1a).

2. Figure S3f, it is better to show quantification of two replicates, MMA signal versus Flag.

Response: We thank the reviewer for this suggestion. Accordingly, the quantification of MMA signal versus FLAG is shown in revised Supplementary Fig. 3f. The mono-methylation of METTL14-T, but not R255K-T, was reduced in response to *PRMT1* knockdown.

3. PRMT1-/- show much higher m6A/A ratio than R255K mutant ESCs, which means either PRMT1 is not the major methyltransferase for R255 or R255K mutation has other impact besides blocking methylation. Can authors comment on these two possibilities? In addition, despite significant differences of m6A/A ratio between PRMT1-/- and R255K mutant (3 fold), MeRIP enrichment showed similar decreases (Figure 2f and 4d) for PRMT1-/- and R255K mutant, can authors explain this discrepancy?

Response: We thank the reviewer for the comments. LC-MS/MS provides a global quantification of m⁶A level on polyadenylated RNAs. By contrast, MeRIP-qPCR is an immunoprecipitation-based m⁶A level assessment for specific regions on total RNAs (including non-polyA RNAs), and has relatively limited utility for quantification of m⁶A stoichiometry (Wang, et al. *Nat Chem Biol* 2020: 896-903). We therefore decided that the m⁶A level detected by these two methods could not be compared directly. To further assess the m⁶A level in R255K and *Prmt1*^{-/-}, we carried out additional MeRIP-qPCR assays to analyze additional R255me dependent m⁶A peaks. We found that the methylation level of most of the R255me regulated m⁶A peaks were reduced in *Prmt1*^{-/-}, but to a lesser extent compared to that in R255K (response Fig. 1b), which is in agreement with the LC-MS/MS results.

Considering that *Prmt1*^{-/-} has only a mild effect on global m⁶A methylation levels, and arginine sites can be targeted (even competitively) by multiple PRMTs (Casadio, et al. *Proceedings of the National Academy of Sciences* 2013: 14894-14899; Girardot, et al. *Nucleic Acids Research* 2014: 235-248; Hyllus, et al. *Genes Dev* 2007: 3369-80; Strahl, et al. *Current Biology* 2001: 996-1000; Zheng, et al. *Molecular Cell* 2013: 37-51), it's possible that other members of the PRMT family are redundantly involved in METTL14 R255 methylation. Meanwhile, PRMT1 is responsible for the arginine methylations for many proteins such as histones (Strahl, et al. *Current Biology* 2001: 996-1000; Wang, et al. *Science* 2001: 853-7), and it is also possible that *Prmt1*^{-/-} could affect the m⁶A level through other indirect pathways. Nevertheless, our results show that PRMT1 interacts with METTL14 (revised Fig. 3a-e and supplementary Fig. 3b-e) and methylates R255 *in vitro* and in cells (revised Fig. 3f and supplementary 3f). Further, while this study was under revision, a new publication showed that PRMT1 mediates the arginine methylation of METTL14 to regulate the interaction between METTL14 and RNA, as similarly concluded in this study (Wang, et al. *Embo j* 2021: e106309). These results suggest that PRMT1 is important for METTL14 methylation.

As for the R255K mutation, K is often used as an unmethylated mimetic of R, because it mimics the positive charge but cannot be methylated by PRMTs (Wei, et al. *Proceedings of the National Academy of Sciences of the United States of America* 2013: 13516-13521; Zheng, et al. *Molecular Cell* 2013: 37-51; Zhu, et al. *Blood* 2019: 1257-1268). Although we can't completely exclude the possibility that the R255K mutation has other impact besides blocking methylation, our results showing that the global m⁶A level were reduced in *Prmt1* knockdown or knockout cells (Fig. 4b-c), methylation levels of R255me regulated m⁶A regions were also reduced in *Prmt1*^{-/-} cells (Fig. 4d), and overexpression of PRMT1 in R255K mESCs could not rescue the decreased mRNA m⁶A level (Fig. 4e), indicate that the methylation of R255 interferes with m⁶A deposition on RNAs.

Overall, we agree that it is possible PRMT1 is not the sole methyltransferase for

R255, and *Prmt1*^{-/-}/R255K may have other impacts besides changing the R255 methylation. In the future, we aim to study the contribution and competition of different PRMTs for METTL14 R255 methylation *in vivo* and during developmental processes, as well as the function of R255 methylation by removing all the side effects of *Prmt1*^{-/-} and R255K, but these are beyond the scope of the current work.

4. Figure 5e, please label genes in each category (m6A decreased, m6A nondownregulated).

Response: We thank the reviewer for this comment, and we have labelled the gene categories in the revised Fig. 5e accordingly.

5. Figure 6j-k, the authors have tried very hard on the mechanistic explanation for defects in endoderm differentiation. However, these experiments are not conclusive and poorly carried out. First, genes selected for knocking down (*Smad7* and *Klf4*) are not even upregulated in R255K mutant; Second, Only *Klf4* is knocked down. I suggest the authors removing this part to avoid confusing future readers.

Response: We thank the reviewer for this suggestion. We have removed this part accordingly.

6. For newly added rescue experiments (Figure 2c and 4c), rescue level for PRMT1 and *Mettl14* should be compared to the level of these genes in WT ESCs, preferably by Western blotting.

Response: We thank the reviewer for this comment. Accordingly, we have shown the protein level of METTL14 and PRMT1 in revised Supplementary Fig. 2d and 4f.

7. Figure 2a, quantification for three replicates should be added. Are these technical replicates or independent experiments?

Response: We thank the reviewer for this comment. Accordingly, we have shown the quantification for replicates in revised Fig. 2a. The replicates were independent experiments.

Response to Reviewer #3

Upon receipt of the reviewer's criticisms, the authors have thoroughly revised their manuscript including new data as well as additional details into the methodology sections. This provides improved confidence for the characterisation of this posttranslational modification in METTL14.

Although the essential information is now provided, including public data repository, some of the newly added sections need further editing to be grammatically correct. Below are a couple of examples:

Response: We thank the reviewer for the positive comments that “*This provides improved confidence for the characterization of this posttranslational modification in METTL14*”. We apologize for the grammatical mistakes and have carefully corrected them in the revised manuscript in the “LC-MS/MS analysis of protein” and “Data availability” method sections. These changes are marked in red.

Minor comments

Line 675: "Mascot search engines" Please provide details, such as version, in-house or public etc.

Response: We thank the reviewer for this comment. The Mascot search engines was Proteome Discoverer 1.3 built-in version v2.3.0. We have added this information in the “LC-MS/MS analysis of protein” section.

Line 684: For the identification of METTL14 interacting proteins....

Response: This has been corrected in revised line 683.

Line 688: ..and a Dionex Ultimate..

Response: This has been corrected in revised line 687.

Lines 689/699: "The peptides were subjected to NSI source" - what does this mean? Please clarify. Something along the lines of:

The peptides were subjected to electrospray ionisation ESI and analysis by tandem mass spectrometry (MS/MS) using an Orbitrap Q Exactive tandem mass spectrometer (ThermoFisher).

Response: We thank the reviewer for pointing this out and have clarified accordingly in revised line 688-690 as follows: “The peptides were then subjected to nanospray ionization (NSI) (ThermoFisher) and analyzed by tandem mass spectrometry (MS/MS) using an Orbitrap Q Exactive tandem mass spectrometer (ThermoFisher)”.

Line 691: Please specify NCE - collision energy

Response: We thank the reviewer for this comment and now specify NCE as “normalized collision energy” accordingly in revised line 691.

Line 694: Mascot search engines - please specify as described above.

Response: We have added the version and details in revised lines 695.

Lines 862/863: The mass spectrometry data has been deposited in integrated proteome resources, such as and

Response: We thank the reviewer for this comment and have clarified in revised lines

884-886 as follow: “The mass spectrometry data has been deposited in integrated proteome resources (iProX) under accession number IPX0002127000 and ProteomeXchange Consortium under ID PXD018458”.

Response to Reviewer #4

I have been specifically asked to concentrate on the modeling and simulation aspects of the manuscript. My expertise lies far from the experimental biochemistry which constitutes the bulk of this work so I will abstain from commenting on that. The simulations were explicitly requested by one of the reviewers and are specifically aimed at clarifying the effect of R255 methylation of METTL14 on the binding of the METTL3-METTL14 complex to RNA substrates. The authors address this issue by performing molecular dynamics simulations of the methylated and unmethylated METTL3-METTL14 complex bound to a GGACU RNA ligand. Results are presented basically in the form of three observables followed in the course of the two simulations (methylated and unmethylated): 1) RMSF of the RNA nucleotides; 2) Number of protein-RNA hydrogen bonds; 3) Binding energy (unspecified, see below) between protein and RNA. The raw data used to generate the plots is also supplied in the form of a separate spreadsheet file, I found this quite helpful.

The idea of measuring the difference in stability of the methylated vs unmethylated systems from equilibrium simulations is quite reasonable. Unfortunately, the data presented are simply insufficient to draw meaningful conclusions (50ns total simulation time per system). This type of analysis generally requires much more sampling of conformational space, with several independent simulation runs. Ideally, to fully resolve such a question, a full-fledged binding free energy (not just enthalpy) study would be necessary. But I understand that this would constitute a new research project on its own and definitely would lie beyond the scope of the present study. Here, it would suffice to run a few more replicas of these simulations, to check that the results are not just a statistical accident. This should not be too problematic to run with YASARA on consumer grade hardware.

The main conclusion of greater stability of the methylated complex derived from the simulations rests on results presented in lines 205-209 of the manuscript: 1) the increase in the number of hydrogen bonds and concomitant decrease in the binding energy and 2) the lower RMSF values of the RNA nucleotides observed upon methylation of R255. The first are only seen in the last 20ns of the simulations, this is really too short a time and the relative difference is too small to exclude that these are not just an effect of statistical fluctuations. For the lower RMSF values it is not clear whether these are calculated over the entire 50ns or simply the last 20ns as above (see minor comments below). In either case, while the trend does point to a greater stability of the methylated complex, this could also be an effect of a bad or not well equilibrated

starting structure. Only additional replicated simulations can help clarify this.

Response: We thank the reviewer for the professional comments and suggestions. Accordingly, we have provided two additional replicated simulations, and similar results were obtained (revised supplementary Fig. 6). Additionally, the RMSF values were calculated over the entire 50 ns.

Additionally, I have a some concerns regarding the simulations presented here:

Major:

The methods section is inadequate. For some of the following issues a bit of clarification is sufficient, but in order to allow others to reproduce/understand what was done, a lot more information needs to be specified:

Response: We thank the reviewer for this constructive comment. Accordingly, we have carefully revised the methods section (see “Molecular simulation” section).

- What is the final size of the simulated system in number of atoms including water, salt, and counter-ions?

Response: We thank the reviewer for this comment. A total of 55599 and 55601 atoms were added to the R255 unmethylated and methylated simulated systems, respectively. We have now added this information in the “Molecular simulation” section of the methods.

- Why was the RNA ligand first equilibrated before docking (Lines 798-801)? Was HDOC used only for rigid body docking?

Response: We thank the reviewer for this comment. The first equilibration and simulation were performed to optimize the initial RNA structure. HDOCK was used for semi-flexible docking. The receptor protein was set as rigid and the ligand RNA was set as flexible.

- What options were chosen in HDOC and what final scores were obtained? How were the docked structures selected? Is the large difference in the RNA structures shown in Fig S5g the result of the docking procedure, the H-bond optimization step, or the MD simulations? It makes a big difference if the starting points for the MD simulations are very similar and then diverge during the simulation, or if the simulations simply started from very different RNA conformations.

Response: We thank the reviewer for this comment. The options were default in HDOCK (Yan, et al. *Nucleic Acids Research* 2017: W365-W373). The structure with top docking score (-210.90) in HDOCK was selected for further analysis. The difference in the RNA structure in Supplementary Fig. 5g was the result of the MD simulation. The simulations were started from the same RNA conformation.

- The simulation protocol is poorly described, what forcefield parameters were used for the methylated R255? What was the equilibration protocol (were any restraints used, for how long)? The authors need to clearly state how long was the full equilibration time vs the production time.

Response: We thank the reviewer for this comment. We have carefully revised the simulation protocol accordingly (see “Molecular simulation” section in the methods). The force field parameters for the methylated R255 were automatically generated by the AutoSMILES procedure in YASARA, and the detailed procedure is shown in response Fig. 2 (Jakalian, et al. *J Comput Chem* 2002: 1623-41; Klamt *The Journal of Physical Chemistry* 1995: 2224-2235; Stewart *Journal of Computer-Aided Molecular Design* 1990: 1-103; Wang, et al. *Journal of Computational Chemistry* 2004: 1157-1174).

The system was equilibrated for 10 ns, and during the equilibration procedure, the coordinates of the backbone chain atoms of protein and RNA were restrained, and the amino acid side-chain atoms were sufficiently released. Then a 50 ns production simulation was carried out.

- What were the criteria for deciding that the system had reached equilibrium? The authors mention RMSD, Rg, and SAS but do not specify how they chose 30ns to start production rather than some other time point. The RMSD, Rg and SAS data should also be supplied as separate plots (my suggestion would be to aggregate all simulation and modeling data in a separate figure).

Response: We thank the reviewer for this comment. RMSD, Rg, and SASA were used to evaluate if the system had reached equilibrium (revised supplementary Fig. 6a-d). After 30 ns, the system came to a relative steady state, so we chose 30-50 ns to calculate the hydrogen bond and binding energy. As per the suggestion, we have supplied the RMSD, Rg and SAS data and aggregated all simulation and modeling data in a separate figure (revised supplementary Fig. 6).

- What pKa prediction algorithm was used to determine protonation states and with what parameters?

Response: We thank the reviewer for this comment. YASARA adopts a fast empirical pKa prediction algorithm (Krieger, et al. *Journal of Molecular Graphics and Modelling* 2006: 481 - 486), which is a function of electrostatic potential, hydrogen bonds and accessible surface area:

$$pK_a = \text{Model } pK_a + \sum_{\text{Ionizable atoms}} [-A \times \text{Ewald}E_i + B \times \text{HB}_i]$$

$$+Sign(HB_{Sum}) \times C \times SurfaceLoss$$

In this equation, Model pK_a is the standard pK_a value of a residue type, A, B and C are proportionality constraints, which are derived from 217 experimentally determined pK_a s. E_{i} represents the reciprocal space portion of the Ewald energy of a charge +1 at the i th ionizable atom. HB_i represents the difference between donated and accepted hydrogen bonds at the i th atom, and HB_{Sum} is the sum of all HB_i . The $SurfaceLoss$ is the loss of accessible surface area of the sidechain compared to a fully exposed state (Krieger, et al. *Journal of Molecular Graphics and Modelling* 2006: 481 - 486).

- How was the binding energy computed? Is this just derived from the intermolecular forcefield terms? Does this adequately represent the binding enthalpy? What about entropic contributions to binding?

Response: We thank the reviewer for this comment. The binding energy was calculated by YASARA through the following equation:

$$E_{bind} = E_{pot_{complex}} + E_{sol_{complex}} - E_{pot_{METTL14}} - E_{sol_{METTL14}} - E_{pot_{RNA}} - E_{sol_{RNA}}$$

where $E_{pot_{complex}}$, $E_{pot_{METTL14}}$ and $E_{pot_{RNA}}$ are the potential energy for the complex, METTL14 and RNA, respectively. The solvent energy component was defined by:

$$E_{sol} = E_{sol_{coulomb}} + E_{sol_{vdw}} + molsurf \times surfcost$$

where $E_{sol_{coulomb}}$ and $E_{sol_{vdw}}$ are the coulomb and Van der Waals components of solvation energy, $molsurf$ is molecular solvent accessible surface areas, $surfcost$ is a guesstimate of the entropic cost of exposing an area in the surface to the solvent in kJ/mol and 0.65 was utilized (<http://www.yasara.org>).

The energy calculation considers the force field, electrostatic and Van der Waals solvation energies, and can adequately represent the binding enthalpy. The methods ignore some entropic components, and thus not are suitable for absolute energy estimation. Nevertheless, the binding energy obtained can be a helpful qualitative estimate to compare the effect of single site methylation on binding. The binding affinity change after R255 methylation was further confirmed by pull down and PAR-CLIP-qPCR experiments (revised Fig.5e-f).

Minor:

Line 194: “simulated” not “stimulated”

Response: We apologize for this mistake and have corrected it.

Line 199-204: Was RMSF calculated only during the equilibration phase and not during the “production phase of the simulations”? This seems unlikely to me, but the

sentence starting at line 201 “We next simulated...” seems to imply simulation was carried out after calculation of RMSF? Please clarify the order of events here.

Response: We thank the reviewer for this comment. The RMSF was calculated during the production phase of the simulations after the equilibration phase. We apologize for this mistake and accordingly we have moved the “We next simulated ...” sentence to the position before RMSF calculation.

Use of units should be precise and consistent. The salt concentration used in the simulations is described in Line 806 as 0.9% NaCl. Is this molar percentage, mass percentage, volume percentage...? My suggestion is to use molar concentrations as one would for any experimental buffer solution. Distances: cutoffs are expressed in Å on Line 810, and in nm on Line 811, please choose one of the two.

Response: We thank the reviewer for this comment. 0.9% NaCl is the mass percentage, and the molar concentration of NaCl is 0.1538 mol/L. The cutoffs are expressed in nm in the revised method as per the suggestion.

In Fig S5g the choice of colors makes it very hard to distinguish the RNA from the protein background, would it be possible to change the color scheme (maybe also highlight the methylation point) to facilitate the reader?

Response: We thank the reviewer for this comment, and we have revised the figure accordingly (revised supplementary Fig.6h).

Line 796: Who or what is Modekeji?

Response: We thank the reviewer for this comment. Modekeji is the corporation which helped us carrying out this simulation.

Response Figures

a

b

Response Fig. 1

- The western blot band intensity of WTAP and METTL3 relative to GAPDH before and after *PRMT1* knockdown. Data are mean \pm s.d. from four independent experiments.
- MeRIP-qPCR of R255 methylation dependent m⁶A peaks in WT, R255K and *Prmt1*^{-/-}. Enrichment of MeRIP versus input RNA was normalized against *Gapdh*. Data are mean \pm s.d. of three independent experiments.

Response Fig. 2

- a. The steps for automatic force field parameter assignment for organic molecules using YASARA AutoSMILES (adapted from the figure in <http://www.yasara.org/autosmiles.htm>).

References

1. Casadio, Fabio, et al. (2013) H3R42me2a is a histone modification with positive transcriptional effects. *Proceedings of the National Academy of Sciences* 110(37):14894-14899.
2. Girardot, Michael, et al. (2014) PRMT5-mediated histone H4 arginine-3 symmetrical dimethylation marks chromatin at G + C-rich regions of the mouse genome. *Nucleic Acids Research* 42(1):235-248.
3. Hyllus, D., et al. (2007) PRMT6-mediated methylation of R2 in histone H3 antagonizes H3 K4 trimethylation. *Genes Dev* 21(24):3369-80.
4. Jakalian, A., D. B. Jack, and C. I. Bayly (2002) Fast, efficient generation of high-quality atomic charges. AM1-BCC model: II. Parameterization and validation. *J Comput Chem* 23(16):1623-41.

5. Klamt, Andreas (1995) Conductor-like Screening Model for Real Solvents: A New Approach to the Quantitative Calculation of Solvation Phenomena. *The Journal of Physical Chemistry* 99(7):2224-2235.
6. Krieger, E., et al. (2006) Fast empirical pKa prediction by Ewald summation. *Journal of Molecular Graphics and Modelling* 25:481 - 486.
7. Stewart, J. J. (1990) MOPAC: A semiempirical molecular orbital program. *Journal of Computer-Aided Molecular Design* 4:1-103.
8. Strahl, Brian D., et al. (2001) Methylation of histone H4 at arginine 3 occurs in vivo and is mediated by the nuclear receptor coactivator PRMT1. *Current Biology* 11(12):996-1000.
9. Wang, H., et al. (2001) Methylation of histone H4 at arginine 3 facilitating transcriptional activation by nuclear hormone receptor. *Science* 293(5531):853-7.
10. Wang, Junmei, et al. (2004) Development and testing of a general amber force field. *Journal of Computational Chemistry* 25(9):1157-1174.
11. Wang, Y., et al. (2020) Antibody-free enzyme-assisted chemical approach for detection of N(6)-methyladenosine. *Nat Chem Biol* 16(8):896-903.
12. Wang, Z., et al. (2021) m(6) A deposition is regulated by PRMT1-mediated arginine methylation of METTL14 in its disordered C-terminal region. *Embo j*:e106309.
13. Wei, Han, et al. (2013) PRMT5 dimethylates R30 of the p65 subunit to activate NF- κ B. *Proceedings of the National Academy of Sciences of the United States of America* 110(33):13516-13521.
14. Yan, Yumeng, et al. (2017) HDOCK: a web server for protein–protein and protein–DNA/RNA docking based on a hybrid strategy. *Nucleic Acids Research* 45(W1):W365-W373.
15. Zheng, S. S., et al. (2013) Arginine Methylation-Dependent Reader-Writer Interplay Governs Growth Control by E2F-1. *Molecular Cell* 52(1):37-51.
16. Zhu, Y., et al. (2019) Targeting PRMT1-mediated FLT3 methylation disrupts maintenance of MLL-rearranged acute lymphoblastic leukemia. *Blood* 134(15):1257-1268.

REVIEWERS' COMMENTS

Reviewer #2 (Remarks to the Author):

I have no further comments. The manuscript is acceptable.

Reviewer #4 (Remarks to the Author):

The authors have competently answered all of my questions and clarified my previous doubts. I believe the MD section is now greatly improved and can be published in its current form.